# Microglia replacement by ER-Hoxb8 conditionally immortalized macrophages provides insight into Aicardi–Goutières syndrome neuropathology

Kelsey M Nemec[1,2], Genevieve Uy[1,3], V Sai Chaluvadi[1,2], Freddy S Purnell[4,5], Bilal Elfayoumi[1], Leila Byerly[6], Micaela L O'Reilly[3], Carleigh A O'Brien[1], William H Aisenberg[1], Sonia I Lombroso[1,7], Xinfeng Guo[8], Niklas Blank[9,10], Chet Huan Oon[1], Fazeela Yaqoob[1], Brian Temsamrit[3], Priyanka Rawat[3], Christoph A Thaiss[9], Will Bailis[11,12], Adam P Williamson[6], Qingde Wang[8], Mariko L Bennett[3,13]*, F Chris Bennett[1,3]*

[1]Department of Psychiatry, Perelman School of Medicine, University of Pennsylvania, Philadelphia, United States; [2]Department of Neuroscience, Perelman School of Medicine, University of Pennsylvania, Philadelphia, United States; [3]Division of Neurology, Children's Hospital of Philadelphia, Philadelphia, United States; [4]Department of Biology, School of Arts and Sciences, University of Pennsylvania, Philadelphia, United States; [5]Epigenetics Institute, University of Pennsylvania, Perelman School of Medicine, Philadelphia, United States; [6]Department of Biology, Bryn Mawr College, Bryn Mawr, United States; [7]Department of Systems Pharmacology and Translational Therapeutics, University of Pennsylvania, Philadelphia, United States; [8]Department of Surgery, University of Pittsburgh School of Medicine, Pittsburgh, United States; [9]Department of Microbiology, Perelman School of Medicine, University of Pennsylvania, Philadelphia, United States; [10]Faculty of Biology, University of Freiburg, Freiburg im Breisgau, Germany; [11]Department of Pathology and Laboratory Medicine, Children's Hospital of Philadelphia, Philadelphia, United States; [12]Department of Pathology and Laboratory Medicine, Perelman School of Medicine, University of Pennsylvania, Philadelphia, United States; [13]Department of Neurology, Perelman School of Medicine, University of Pennsylvania, Philadelphia, United States

*For correspondence:
bennettm2@chop.edu (MLB);
frederick.bennett@
pennmedicine.upenn.edu (FCB)

**Competing interest:** The authors declare that no competing interests exist.

## eLife Assessment

This revised study describes an **important** new model for in vivo manipulation of microglia, exploring how mutations in the Adar1 gene within microglia contribute to Aicardi-Goutières Syndome. The methodology is validated with **exceptional** data, supporting the authors' conclusions. The paper underscores both the advantages and limitations of using transplanted cells as a surrogate for microglia, making it a resource that is of value for biologists studying macrophages and microglia.

**Abstract** Microglia, the brain's resident macrophages, can be reconstituted by surrogate cells – a process termed 'microglia replacement'. To expand the microglia replacement toolkit, we here introduce estrogen-regulated (ER) homeobox B8 (Hoxb8) conditionally immortalized macrophages,

a cell model for generation of immune cells from murine bone marrow, as a versatile model for microglia replacement. We find that ER-Hoxb8 macrophages are highly comparable to primary bone marrow-derived macrophages in vitro, and, when transplanted into a microglia-free brain, engraft the parenchyma and differentiate into microglia-like cells. Furthermore, ER-Hoxb8 progenitors are readily transducible by virus and easily stored as stable, genetically manipulated cell lines. As a demonstration of this system's power for studying the effects of disease mutations on microglia in vivo, we created stable, Adar1-mutated ER-Hoxb8 lines using CRISPR-Cas9 to study the intrinsic contribution of macrophages to Aicardi–Goutières syndrome (AGS), an inherited interferonopathy that primarily affects the brain and immune system. We find that Adar1 knockout elicited interferon secretion and impaired macrophage production in vitro, while preventing brain macrophage engraftment in vivo – phenotypes that can be rescued with concurrent mutation of Ifih1 (MDA5) in vitro, but not in vivo. Lastly, we extended these findings by generating ER-Hoxb8 progenitors from mice harboring a patient-specific Adar1 mutation (D1113H). We demonstrated the ability of microglia-specific D1113H mutation to drive interferon production in vivo, suggesting microglia drive AGS neuropathology. In sum, we introduce the ER-Hoxb8 approach to model microglia replacement and use it to clarify macrophage contributions to AGS.

## Introduction

Microglia, the parenchymal tissue-resident macrophages of the brain and spinal cord, play critical roles in development, homeostasis, injury, and disease (*Salter and Stevens, 2017*; *Li and Barres, 2018*). When depleted of endogenous microglia, the brain parenchyma can be reconstituted by surrogate macrophages – a process termed 'microglia replacement'. Microglia replacement has the potential for precise and personalized delivery of therapeutic payloads or correction of dysfunction. Primary microglia are challenging to manipulate – they rapidly lose transcriptional identity ex vivo, resist viral manipulation, are minimally proliferative, and are highly sensitive to serum (*Balcaitis et al., 2005*; *Jiang et al., 2012*; *Masuda et al., 2013*; *Su et al., 2016*; *Bohlen et al., 2017*). These challenges limit the study of microglia replacement, underscoring the need for new, transplantable cell models. Common microglia surrogate cells include myeloid cells from the blood or bone marrow and induced pluripotent stem cell (iPSC)-derived microglia (iMG; *Muffat et al., 2016*; *Pandya et al., 2017*; *Haenseler et al., 2017*; *Douvaras et al., 2017*; *Abud et al., 2017*; *Takata et al., 2020*). Although capable of reconstituting the microglial niche (*Priller et al., 2001*; *Bohlen et al., 2017*; *Bennett et al., 2018*; *Hasselmann et al., 2019*), primary cells and iPSCs each have limitations that motivated us to consider new microglia replacement tools.

To create an alternative microglia surrogate, we turned to the estrogen-regulated (ER) homeobox B8 (Hoxb8) system, a method for generating and gene-editing unlimited quantities of macrophages from primary murine bone marrow (*Wang et al., 2006*). When overexpressed, Hoxb8 promotes the expansion of hematopoietic progenitor cells while preventing their differentiation. When transduced with ER-Hoxb8 virus, myeloid progenitor cells from murine bone marrow become immortalized and indefinitely expandable when in the presence of exogenous estrogen, but differentiate upon estrogen removal. The ER-Hoxb8 approach has been used to generate so-called 'conditionally immortalized' progenitors with lymphoid and myeloid potential (*Wang et al., 2006*; *Rosas et al., 2011*; *Redecke et al., 2013*; *Gurzeler et al., 2013*; *Zach et al., 2015*; *Fites et al., 2018*; *Ma et al., 2020*; *Lail et al., 2022*), including macrophages.

We recently showed that ER-Hoxb8 cells can reconstitute the microglial niche (*Chadarevian et al., 2023*), supporting their potential as a new cell model for microglia replacement. Here, we aimed to deeply characterize the identity, function, and application of ER-Hoxb8s as microglia surrogates to test hypotheses about microglia-specific gene functions in health and disease.

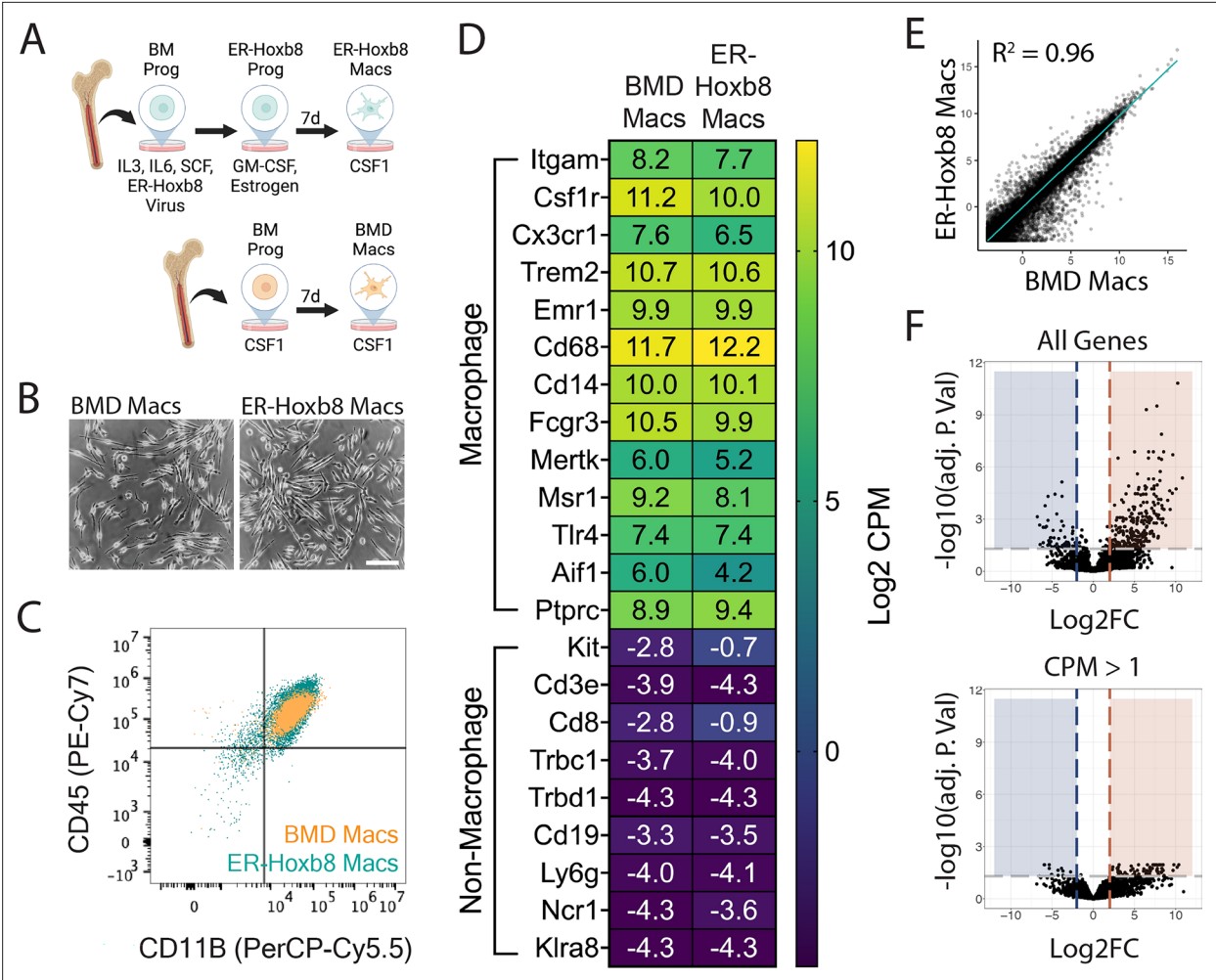

**Figure 1.** Comparison of ER-Hoxb8 to bone marrow-derived (BMD) macrophages in vitro. (**A**) Schematic for creation of BMD and ER-Hoxb8 cells. (**B**) Brightfield images of BMD and ER-Hoxb8 macrophages plated in the presence of 30 ng/mL mouse CSF1 and differentiated for 7 days (scale bar = 100 um). (**C**) Dot plot representing CD45/CD11B levels (pre-gated on live, singlet, leukocyte) by flow cytometry. (**D**) Heatmap showing Log2 CPM of canonical macrophage (top) and non-macrophage (bottom) immune cell genes. (**E**) Whole-transcriptome comparison between BMD and ER-Hoxb8 macrophages, depicting best fit line and coefficient of determination (one dot = one gene). (**F**) Volcano plots comparing all genes or those with CPM >1 (Log2FC ≥ 2, FDR <0.05); blue = upregulated in ER-Hoxb8 macrophages, red = upregulated in BMD macrophages. Panel (**A**) created with https://Biorender.com/w42j197.

The online version of this article includes the following source data and figure supplement(s) for figure 1:

**Source data 1.** Raw numerical values for *Figure 1* plots.

**Figure supplement 1.** Extended validation and comparison of ER-Hoxb8 to bone marrow-derived (BMD) macrophages in vitro, relating to *Figure 1*.

**Figure supplement 1—source data 1.** Raw numerical values for *Figure 1—figure supplement 1* plots.

## Results

### Comparison of ER-Hoxb8 to bone marrow-derived macrophages in vitro

To directly compare primary bone marrow-derived (BMD) and ER-Hoxb8 cells, we created three biologically independent lines of conditionally immortalized macrophage progenitors as described previously (*Wang et al., 2006*). ER-Hoxb8 progenitor cells were plated, differentiated for 7 days in CSF1, and compared to BMD macrophages (*Stanley and Heard, 1977*; *Murray et al., 2014*; *Figure 1A*). We found that ER-Hoxb8 macrophages were highly similar to BMD macrophages morphologically (*Figure 1B*) and by CD11B/CD45 expression (*Figure 1C*, *Figure 1—figure supplement 1A/B*). To validate ER-Hoxb8 macrophage function, we tested their phagocytic capacity. In agreement with prior

literature (*Elhag et al., 2021*), ER-Hoxb8 macrophages more efficiently engulfed beads coated with 10% phosphatidlyserine (PS) to mimic apoptotic cells, compared to beads coated with phosphatidyl-choline (PC) alone (*Figure 1—figure supplement 1C/D*).

To expand upon previous studies (*Redecke et al., 2013*; *Roberts et al., 2019*; *Accarias et al., 2020*), we performed RNA sequencing of both cell types before and after 7 days of differentiation. We generated high-quality transcriptomes (Phred Score >35, n=3 biological replicates) for BM progenitors, BM monocytes, 7-day differentiated BMD macrophages, ER-Hoxb8 progenitors, 4-day differentiated ER-Hoxb8 cells, and 7-day differentiated ER-Hoxb8 macrophages (top 100 expressed genes in *Supplementary file 1A*). In agreement with previous literature (*Wang et al., 2006*), ER-Hoxb8 progenitors express canonical macrophage genes (*Figure 1—figure supplement 1E*) and, interestingly, ER-Hoxb8 cells at all time points were more similar to BMD macrophages than to BM progenitors or monocytes (*Figure 1—figure supplement 1F*).

Transcriptionally, 7-day differentiated ER-Hoxb8 macrophages were highly similar to BMD macrophages as demonstrated by expression of macrophage-specific genes and lack of expression of common progenitor, lymphocyte, neutrophil, and natural killer cell-specific genes (*Figure 1D*). Linear regression analysis between the two cell types (*Figure 1E*) revealed a strong correlation ($R^2$=0.96) between expression levels of all 29,625 expressed genes. We identified 85 genes as differentially

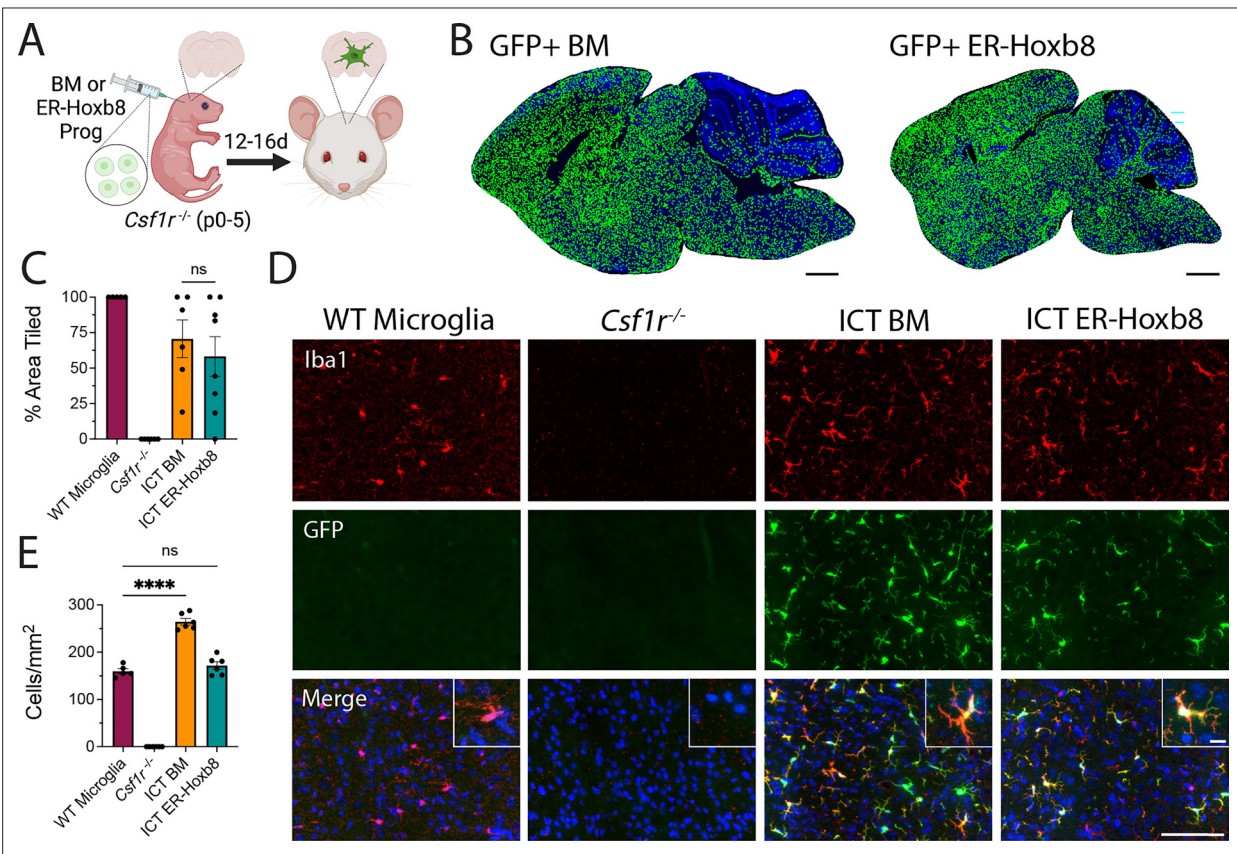

**Figure 2.** Engraftment potential of ER-Hoxb8 compared to bone marrow-derived (BMD) macrophages after intracranial transplantation in Csf1r$^{-/-}$ hosts. (**A**) Schematic for in vivo Csf1r$^{-/-}$ transplant experiments. (**B**) Rendered tile stitches of Csf1r$^{-/-}$ brains after intracranial injection of GFP+ bone marrow (left) or ER-Hoxb8 (right) progenitor cells. (**C**) Percent of total brain area tiled by donor cells; n=5–7 biological replicates per group; each dot = one biological replicate (average area across three matched sagittal sections). (**D**) Immunostaining of cortical brain region 12–16 days post-intracranial injection (red = IBA1, green = endogenous GFP, blue = DAPI; scale bar = 100 um; inset scale bar = 5 um). (**E**) Cortical density calculations (cells per mm$^2$) between groups; n=5–7 biological replicates per group; each dot = one biological replicate (average density across three regions of interest across three matched sagittal sections). All p-values calculated via one-way ANOVA with multiple comparisons; ns = not significant or p≥0.05, ****p<0.0001. Panel (**A**) created with https://BioRender.com/j85e198.

The online version of this article includes the following source data for figure 2:

**Source data 1.** Raw numerical values for *Figure 2* plots.

expressed between ER-Hoxb8 and BMD macrophages (FDR <0.05, Log2 fold change ≥ 2, counts per million [CPM] >1; *Figure 1F*). Of these, 74 were upregulated in BMD macrophages, 11 were upregulated in ER-Hoxb8 macrophages, and no statistically significant Gene Ontology (GO) terms were found for either group (top 10 DEGs by Log2FC in *Figure 1—figure supplement 1G*, all DEGs in *Supplementary file 1B*). These data demonstrate that ER-Hoxb8 macrophages are highly similar to primary BMD macrophages in vitro.

## ER-Hoxb8 cells engraft the brain parenchyma after intracranial transplantation and attain a microglia-like identity

To extend the utility of the ER-Hoxb8 model, we next studied their engraftment and identity following intracranial transplantation into Csf1r$^{-/-}$ hosts, which lack microglia and readily permit donor cell engraftment (*Bennett et al., 2018*; *Figure 2A*). Using BMD cells as a primary cell control, early postnatal transplantation at days 0–5 (P0-5) led to robust and comparable parenchymal engraftment by 16 days (*Figure 2B–D*). ER-Hoxb8 macrophages engrafted at equivalent densities (171.7 cells/mm$^2$ ± 8.0) as endogenous, wild-type (WT) microglia (160.0 cells/mm$^2$ ± 5.5), while BMD macrophages engrafted at higher densities (264.6 cells/mm$^2$ ± 6.9; *Figure 2E*).

BMD macrophages become 'microglia-like' after engraftment in the Csf1r$^{-/-}$ brain (*Bennett et al., 2018*). Like BMD cells, ER-Hoxb8 cells lack expression of the microglia signature protein TMEM119

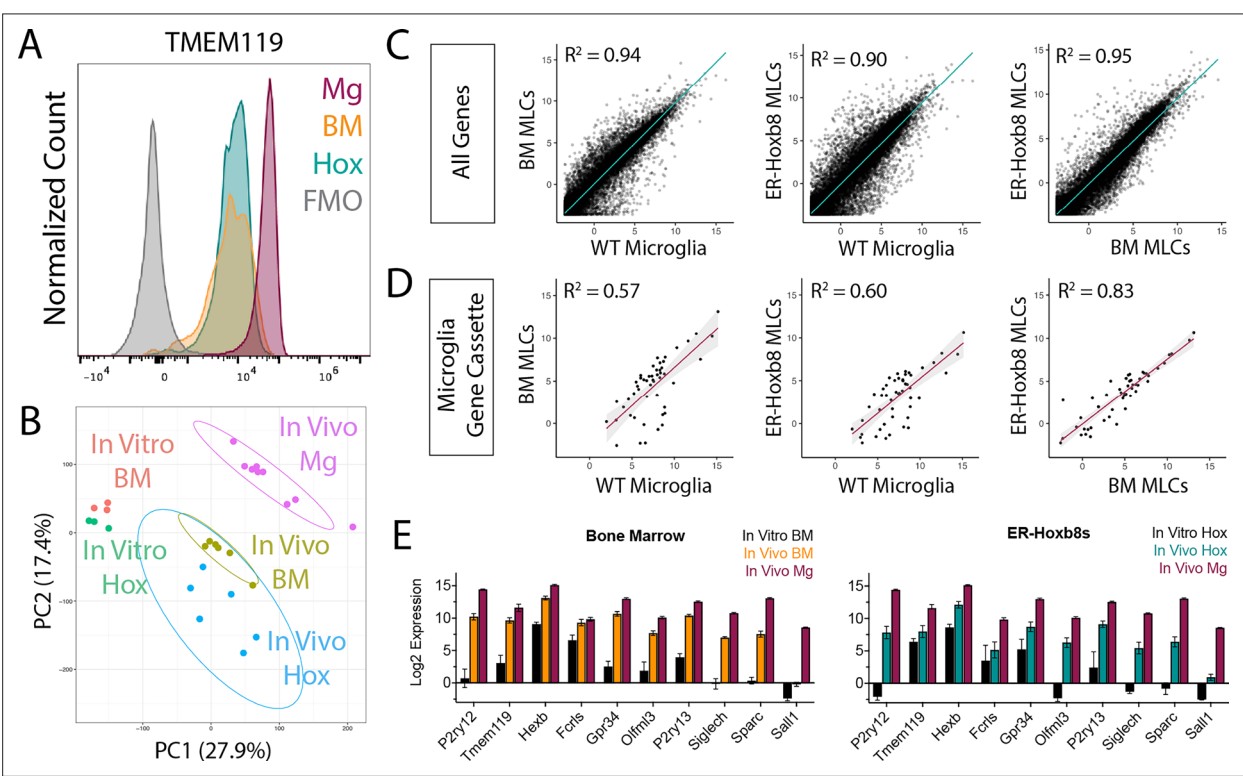

**Figure 3.** ER-Hoxb8 macrophages become microglia-like cells (MLCs) after engraftment in the Csf1r$^{-/-}$ brain. (**A**) Histogram of TMEM119 surface staining by flow cytometry (pre-gated on live, singlet, leukocyte, CD45+/CD11B+) for brain-engrafted cells 14 days post-intracerebral transplantation; Mg = WT Microglia, BM = BMD MLCs, Hox = ER-Hoxb8 MLCs. (**B**) PCA plot comparing in vitro macrophages from *Figure 1* with in vivo macrophages; Mg = WT Microglia, BM = BMD MLCs, Hox = ER-Hoxb8 MLCs. (**C**) Whole-transcriptome comparison between WT microglia, BMD, and ER-Hoxb8 macrophages in vivo, depicting best fit line and the coefficient of determination. (**D**) Comparison of microglia signature genes (*Cronk et al., 2018*) depicting best fit line and coefficient of determination. (**E**) In vitro and in vivo Log2 CPM gene expression of 10 canonical microglia/myeloid genes for bone marrow (left) and ER-Hoxb8s (right) as compared to unmanipulated in vivo microglia.

The online version of this article includes the following source data and figure supplement(s) for figure 3:

**Source data 1.** Raw numerical values for *Figure 3* plots.

**Figure supplement 1.** Extended comparison of ER-Hoxb8 and bone marrow-derived (BMD) macrophages after intracranial transplantation in Csf1r$^{-/-}$ hosts, relating to *Figure 3*.

**Figure supplement 1—source data 1.** Raw numerical values for *Figure 3—figure supplement 1* plots.

in vitro (*Figure 3—figure supplement 1A*). After brain engraftment, we found comparably high levels of TMEM119 in BMD and ER-Hoxb8 macrophages by flow cytometry, though both had lower TMEM119 expression than endogenous microglia, consistent with prior studies (*Bennett et al., 2018*; *Cronk et al., 2018*; *Shemer et al., 2018*; *Figure 3A*). As with WT microglia and BMD macrophages, ER-Hoxb8 macrophages likewise express P2RY12 after brain engraftment (*Figure 3—figure supplement 1B*).

To more comprehensively assess microglial identity, we generated high-quality bulk RNA sequencing transcriptomes (Phred Score >35; n=6–9 biological replicates, *Figure 3—figure supplement 1C*) from isolated brain-engrafted BMD macrophages, ER-Hoxb8 macrophages, and endogenous microglia from age-matched WT hosts. Principal component analysis (PCA) and unsupervised hierarchical clustering revealed that brain residence induces both BMD and ER-Hoxb8 macrophages to become more similar to endogenous microglia (*Figure 3B*, *Figure 3—figure supplement 1D*). We saw no evidence for batch effects between harvest days, cell sorter used, or host mouse sex (*Figure 3—figure supplement 1E*). BMD and ER-Hoxb8 microglia-like cells (MLCs) were transcriptionally similar, by both PCA and linear regression analysis (*Figure 3B/C/D*). ER-Hoxb8 MLCs upregulated expression of canonical microglia signature genes akin to BMD controls (P2ry12, Tmem119, Hexb, Fcrls, Gpr34, Olfml3, P2ry13, Sparc), including genes highly downregulated in vitro (*Figure 3E*). Like BMD macrophages, ER-Hoxb8 macrophages lack Sall1 expression in vivo compared to microglia (*Figure 3E*). Lastly, to more completely assess the degree that ER-Hoxb8 cells become 'microglia-like' after brain engraftment, we compared the top up- and downregulated genes from our datasets to those of brain-engrafted yolk sac (YS) progenitors using data from *Bennett et al., 2018* (*Figure 3—figure supplement 1F*; *Supplementary file 1C*). We find overlap between groups, which likewise provides a better understanding of genes that are robustly regulated by brain engraftment. Taken together, these data show that, similar to BMD macrophages, ER-Hoxb8 macrophages become "microglia-like" with exposure to signals in the brain microenvironment.

Despite their similarities, BMD MLCs and ER-Hoxb8 MLCs have 650 differentially expressed genes (FDR <0.05, Log2FC ≥ 2, and CPM >1 in at least six of the samples (the 'n' of each group); *Supplementary file 1D*). We performed statistical overrepresentation tests to identify relevant GO terms (*Supplementary file 1E*). Of particular interest were terms relating to biological processes as they concerned immune activity and reactivity. We discovered that the 34 overrepresented genes for these GO terms (*Supplementary file 1F*), were unique to ER-Hoxb8 MLCs, suggesting that the brain environment differentially affects engrafted BMD and ER-Hoxb8 cells, and that ER-Hoxb8 MLCs may be more primed to activation.

To gauge how significantly these transcriptomic increases in inflammation-associated genes impact the brain environment, we stained for GFAP, a marker of reactive astrocytosis and a sensitive marker for central nervous system (CNS) perturbations. We found no increase in GFAP expression across brain regions between any groups, including in brains engrafted with ER-Hoxb8 cells compared to BMD cells (*Figure 3—figure supplement 1G*).

## *Adar1* mutation reduces macrophage number and induces interferon responses, effects mitigated by JAK inhibition or Ifih1 mutation

Having established ER-Hoxb8 macrophages as microglia-like cells, we explored their potential for study of microglia in health and disease. Building on prior work (*Gran et al., 2018*; *Roberts et al., 2019*; *Abud et al., 2017*; *Bromberger et al., 2022*; *Shen et al., 2022*; *Xu et al., 2022*; *Möller et al., 2023*), we used CRISPR-Cas9 to knockout (KO) Tlr4 in Cas9[+/-] ER-Hoxb8 progenitors, using fluorescence-activated cell sorting (FACS) to establish lines of guide-transduced ER-Hoxb8 progenitors (*Figure 4—figure supplement 1A*). We confirmed editing by TIDE analysis (*Figure 4—figure supplement 1B*, *Brinkman et al., 2014*) and protein loss by flow cytometry (*Figure 4—figure supplement 1C/D*). We then verified phenotypic knockout by treating unmodified, non-targeting control (NTC) guide-transduced, and Tlr4 guide-transduced ER-Hoxb8 macrophages with lipopolysaccharide (LPS, a Tlr4 agonist) or R848 (a Tlr7/8 agonist). As expected, Tlr4 KO blunted TNF-a production in response to Tlr4 but not Tlr7/8 agonism (*Figure 4—figure supplement 1E*).

Having validated effective gene targeting, we next leveraged strengths of the ER-Hoxb8 system to study the contribution of microglia to a monogenic CNS disease. Aicardi–Goutières syndrome (AGS) is a rare, brain-predominant genetic interferonopathy with nine known causal genes (*Gavazzi et al.,*

*2024*). As many cells respond to interferon, it is unclear which cells drive AGS brain pathology, particularly in cases where global gene knockout is embryonic lethal. We therefore tested the hypothesis that microglia, which produce and react strongly to interferons (*Zheng et al., 2015*; *Aw et al., 2020*; *Escoubas et al., 2024*), are a main contributor to AGS pathology. We first confirmed that the seven coding AGS-causal genes are expressed by ER-Hoxb8 cells at similar levels to those of BMD cells and true microglia (*Figure 4—figure supplement 2A*). We were primarily interested in Adar1, an RNA editing enzyme expressed by all brain cells, including microglia (*Zhang et al., 2014*; *Bennett et al., 2016*). As ADAR1 loss disrupts hematopoiesis (*Arshadi et al., 2021*; *Hartner et al., 2004*), hematopoietic progenitor differentiation (*XuFeng et al., 2009*), and is embryonically lethal in mice (*Arshadi et al., 2021*; *Hartner et al., 2004*), the impact of ADAR1 loss in macrophages is unknown. We leveraged the ER-Hoxb8 model to study Adar1 loss in macrophages by creating independent Adar1 KO lines using two distinct sgRNAs targeting different exons (*Figure 4A*, *Figure 4—figure supplement 2B*).

We first explored differences during differentiation of Adar1 KO ER-Hoxb8 cells in vitro. Despite normal numbers at 3 days, there were significantly fewer Adar1 KO cells during the later stages of differentiation compared to NTC cells (*Figure 4B/C*, *Figure 4—figure supplement 2C/D*). This suggests that either Adar1 KO inhibits progenitor differentiation or macrophage proliferation and survival. We next performed RNA sequencing of Adar1 KO progenitors and differentiated macrophages. Adar1 KO progenitors were highly similar to NTC control progenitors (*Figure 4D/E/F*; all DEGs in *Supplementary file 1G*), and Adar1 KO macrophages expressed comparable levels of macrophage-identity genes to NTC (*Figure 4G*), suggesting that Adar1 KO ER-Hoxb8 progenitors can generate macrophages. At the macrophage stage, however, we found 547 DEGs (*Figure 4D/E/F*; *Supplementary file 1H*), remarkable for interferon-stimulated gene (ISG) upregulation and GO term enrichment for responses to virus, stress, cytokine stimulation, and innate immunity (*Figure 4E/F*, *Figure 4—figure supplement 2E*; *Supplementary file 1I*).

To test whether Adar1 KO-mediated interferon production underlies this macrophage phenotype, we used the Janus kinase (JAK) inhibitor Baricitinib to inhibit interferon signaling. Baricitinib, a treatment for AGS (*Vanderver et al., 2020*; *Han et al., 2022*; *Kanazawa et al., 2023*), caused a dose-dependent reduction in ISG expression (*Figure 4H*) and normalized cell counts during macrophage differentiation (*Figure 4I*, *Figure 4—figure supplement 2F*). We validated increased type I-specific interferon (IFN-b) and ISG (CXCL10, IL6, CCL5) production by Adar1 KO ER-Hoxb8 macrophages, and their suppression by baricitinib treatment using multiplex bead array (*Figure 4J*). As interferons inhibit hematopoiesis and differentiation (*Demerdash et al., 2021*), we confirmed that the baricitinib-treated cells were indeed macrophages (*Figure 4G*) and that the ISG reduction improved macrophage production.

Melanoma differentiation associated protein 5 (MDA5, or Ifih1) is an epistatic modifier of Adar1 such that deletion rescues some Adar1 phenotypes (*Liddicoat et al., 2015*; *Guo et al., 2022*). We introduced a guide targeting Ifih1 (BFP+) to Adar1 KO cells (GFP+). We created single-cell clones and validated double KO (dKO, GFP/BFP+) by TIDE analysis (*Figure 4—figure supplement 2H*). Ifih1 loss completely rescued the Adar1 KO cell growth deficit during differentiation (*Figure 4K*), and normalized production of IFN-b, IL-6, and CCL5 (*Figure 4L*), but not CXCL10. As with baricitinib treatment (*Figure 4—figure supplement 2G*), we found that multiple chemokines/cytokines are downregulated in Adar1-mutant cells as compared to NTC (TNF-a, IL-17, MIP-1a, MIP-1B, MIP-2, M-CSF), and are partially rescued by Ifih1 deletion (*Figure 4—figure supplement 2I*).

Together, these data show that Adar1 deletion impairs macrophage but not progenitor health and is associated with interferonopathy, demonstrating the utility of the ER-HoxB8 model system for experimental isolation and study of macrophage dysfunction in a genetic disease.

## *Adar1* mutation prevents ER-Hoxb8 engraftment

We next attempted to study how Adar1 KO macrophages affect the CNS. To our surprise, although TLR4 KO gene edited macrophages engrafted robustly (n=10; *Figure 5A/E*), mice injected with Adar1 KO cells died early and showed no engraftment in the brains of any transplanted mice that survived to endpoint (n=3; *Figure 5B/C*).

Because blocking interferon signaling with baricitinib or Ifih1 deletion correlated with better macrophage survival in vitro, we wondered if modifying our approach to limit interferon tone would

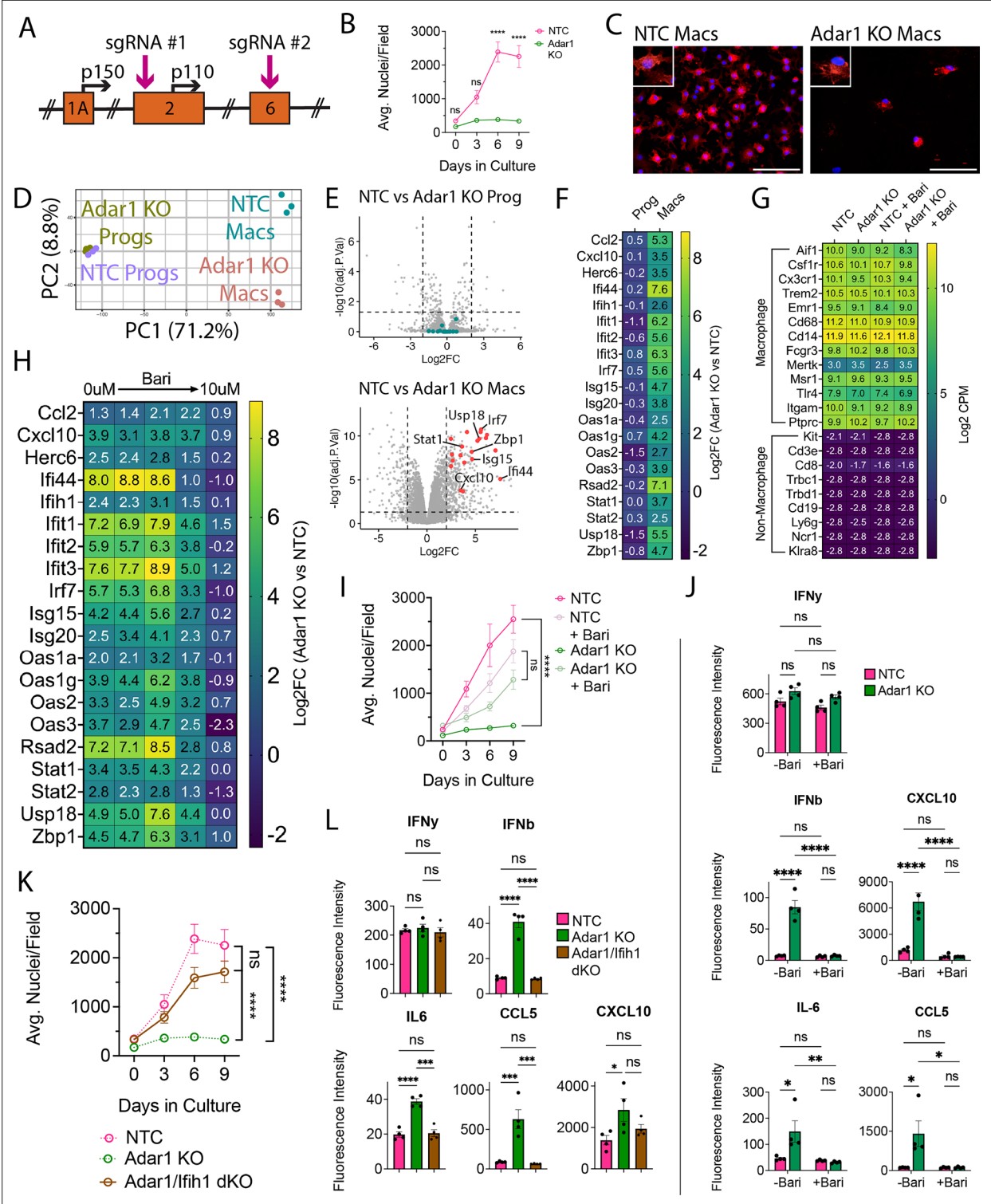

**Figure 4.** Adar1 mutation prevents macrophage-lineage cell expansion and causes interferon induction, rescued by JAKi or Ifih1 mutation. (**A**) Schematic of ADAR1 locus, depicting exons, alternative start sites for p150 and p110 isoforms, and sgRNA targets. (**B**) ER-Hoxb8 cell counts over differentiation time course. (**C**) Immunostaining of in vitro, 8-day differentiated macrophages comparing control (NTC) and Adar1 guide-transduced macrophages (red = CD11B, blue = DAPI; scale bar = 100 um) (**D**) PCA plot of progenitors and macrophages in vitro (**E**) Volcano plots showing differentially expressed genes between Adar1 KO and NTC progenitors and macrophages (CPM >1, Log2FC ≥ 2, FDR <0.05). (**F**) Heatmap showing the Log2FC (Adar1 KO values over NTC values) for relevant interferon-stimulated genes. (**G**) Heatmap showing Log2 CPM of canonical macrophage (top) and non-macrophage (bottom) immune cell genes. (**H**) Heatmap showing Log2FC (Adar1 KO over NTC expression) for interferon-stimulated genes in

*Figure 4 continued on next page*

*Figure 4 continued*

macrophages treated with baricitinib. (**I**) ER-Hoxb8 cell counts over differentiation time course, comparing the effect of baricitinib on Adar KO and NTC lines (dosages = 0 uM, 0.00064 uM, 0.16 uM, 0.4 uM, and 10 uM); statistics calculated at the 9-day time point. (**J**) Interferon, cytokine, and chemokine production after treatment with baricitinib via cytokine bead array. (**K**) ER-Hoxb8 cell counts over differentiation time course, comparing NTC, Adar1 KO, and Adar/Ifih1 double KO (dKO) lines – dotted NTC and Adar1 KO lines are equivalent to those shown in panel (**B**). (**L**) Interferon, cytokine, and chemokine production via cytokine bead array, comparing NTC, Adar KO, and Adar/Ifih1 dKO lines. All p-values calculated via one-way (**L**) or two-way (**B, I, J, K**) ANOVA with multiple comparisons; ns = not significant or p≥0.05, *p<0.05, **p<0.01, ***p<0.001, ****p<0.0001.

The online version of this article includes the following source data and figure supplement(s) for figure 4:

**Source data 1.** Raw numerical values for *Figure 4* plots.

**Figure supplement 1.** CRISPR-Cas9 editing of ER-Hoxb8 progenitors, relating to *Figures 4 and 5*.

**Figure supplement 1—source data 1.** Raw numerical values for *Figure 4—figure supplement 1* plots.

**Figure supplement 2.** Evidence for gene knockout (KO), Adar1 sgRNA #2 results, and extended bead array data, relating to *Figure 4*.

**Figure supplement 2—source data 1.** Raw numerical values for *Figure 4—figure supplement 2* plots.

permit Adar1 KO cell engraftment. We first reduced the number of injected cells (n=4) and harvested tissues earlier, between 4 and 8 days post-transplant (n=8), and did not observe engraftment. We then tried to pre-treat donor cells and host mice with Baricitinib and likewise did not observe engraftment in surviving mice (n=1; *Figure 5E*). Lastly, we transplanted Adar1/Ifih1 dKO cells (n=3), which showed partial rescue of Adar1 KO phenotypes in vitro. Although we detected no areas of engraftment meeting the stringent criteria applied for our 'percent area' quantification method, we noted diffuse patches of engrafted cells with rounded morphology and enlarged soma size, in three of three experimental replicates (*Figure 5D/E*, *Figure 5—figure supplement 1A*), potentially consistent with a mild partial rescue of engraftment.

Csf1r$^{-/-}$ mice have severe constitutional, skeletal, and CNS abnormalities, typically succumbing by weaning age (*Dai et al., 2002*). To extend our findings about Adar1 KO macrophages in vivo, we applied an inducible microglia depletion model using healthy Cx3cr1CreERT; Csf1rfl/fl hosts (*Bennett et al., 2018*). After depleting endogenous microglia via subcutaneous tamoxifen injection at age P1 and P2, we intracerebrally transplanted cells into Cx3cr1CreERT; Csf1rfl/fl brains (*Figure 5—figure supplement 1B*). After 7–15 days in vivo, we saw engraftment of control (TLR4 KO and NTC; n=4; *Figure 5—figure supplement 1C*), but not Adar1 KO cells (n=5; *Figure 5—figure supplement 1D*). Further mirroring the Csf1r$^{-/-}$ data, we saw small, diffuse patches of engrafted Adar1/Ifih1 dKO cells, but none reached quantifiable levels of engraftment (n=11; *Figure 5—figure supplement 1E*). Lastly, we harvested transplanted Cx3cr1CreERT; Csf1rfl/fl hosts early, at day 3, and saw a diffuse pattern of non-quantifiable but parenchymally engrafted Adar1 KO cells (n=4; *Figure 5—figure supplement 1F*). These data suggest that Adar1 KO cells can enter the brain parenchyma but do not persist, consistent with the reduction in cell numbers observed in vitro during macrophage differentiation.

Overall, these results show that genetically modified ER-Hoxb8 cells robustly engraft in the brain parenchyma, which is prevented by Adar1 KO, independent of interferon production.

### *Adar1* D1113H mutant ER-Hoxb8 macrophages drive brain ISG expression

Most available mouse models of AGS are either embryonic lethal (*Arshadi et al., 2021*; *Hartner et al., 2004*; *Liddicoat et al., 2015*) or fail to produce CNS phenotypes (*Morita et al., 2004*; *Ward et al., 2011*; *Hiller et al., 2012*; *Pereira-Lopes et al., 2013*; *Behrendt et al., 2013*; *Rehwinkel et al., 2013*; *Ohto et al., 2022*). We recently created a viable mouse model harboring a patient-derived mutation (D1113H) in the catalytic domain of ADAR1, which displays robust brain ISG expression, astrocytosis, microgliosis, and white matter calcifications (*Guo et al., 2022*). Since AGS patient mutations are typically hypomorphic (rather than knockout), this more authentically models AGS (*Rice et al., 2014*). Given the non-engraftment of Adar1 KO cells, we created ER-Hoxb8 cells from Adar1 D1113H mice to explore the impact of AGS-specific mutations in microglia. Unlike Adar1 KO cells, D1113H ER-Hoxb8s showed normal growth and expansion during in vitro differentiation (*Figure 6A*), despite a similarly increased production of IFN-b, ISGs, and other cytokines, including CXCL10, CXCL1, CCL5, VEGF, MIP-1a, and MIP-1b, but not IFN-g (*Figure 6B*).

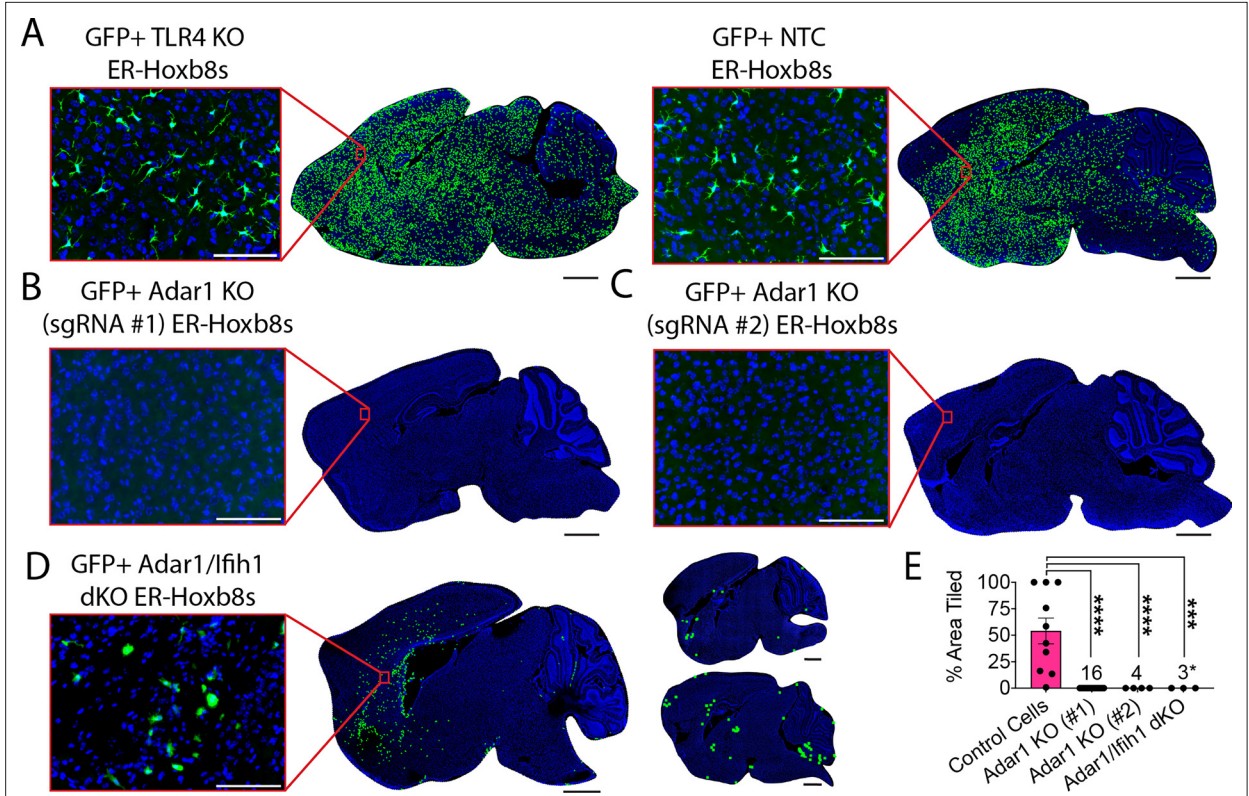

**Figure 5.** Adar1 mutation prevents ER-Hoxb8 engraftment in the Csf1r$^{-/-}$ mouse, partially rescued by Ifih1 deletion. (**A**) Representative rendering of donor cell engraftment (scale bar = 1000 um) with inset microscopy of GFP+ donor cell engraftment (green = endogenous GFP, blue = DAPI; scale bar = 100 um) for control cells (TLR4 KO and NTC) harvested 7–15 days post-injection (dpi), (**B**) Adar1 KO (sgRNA #1) cells (harvest details in **E**), (**C**) Adar1 KO (sgRNA #2) cells harvested 9-12dpi, and (**D**) Adar1/Ifih1 double KO (dKO) cells harvested 10–15 dpi (rendered dots in two right brains enlarged 5x for visualization). (**E**) Percent of total brain area tiled with cells between groups (numbers denote 'n' per group); Adar1 KO (#1) cells include pooled data (brains injected with 300k cells/hemisphere, harvested at 10–15 days post-injection (dpi; n=3); brains injected with 300k cells/hemisphere, harvested at 4–8 dpi (n=8); brains injected with 50k cells/hemisphere, harvested at 13 dpi (n=4); and brains injected with 100k cells/hemisphere pre-treated with 0.5 uM Baricitinib, mice treated daily with 1 mg/kg Baricitinib, harvested at 5 dpi (n=1)); asterisk indicates samples where engraftment is present but does not meet criteria for tiled brain area, as exemplified in (**D**); p-values calculated via one-way ANOVA with multiple comparisons; ns = not significant or p≥ 0.05, ***p<0.001, ****p<0.0001.

The online version of this article includes the following source data and figure supplement(s) for figure 5:

**Source data 1.** Raw numerical values for *Figure 5* plots.

**Figure supplement 1.** Concurrent Adar1, Ifih1 mutation morphology, plus engraftment in the Cx3cr1CreERT; Csf1rfl/fl mouse, relating to *Figure 5*.

**Figure supplement 1—source data 1.** Raw numerical values for *Figure 5—figure supplement 1* plots.

We next characterized D1113H ER-Hoxb8 cells after transplantation into the Cx3cr1CreERT; Csf1rfl/fl brain. As D1113H cells are not genetically tagged, we performed RNA in situ hybridization (ISH), using Cre expression (*Figure 6—figure supplement 1A/B*) to distinguish endogenous microglia (Cre+) from donor cells (Cre-). Unlike Adar KO cells, D1113H mutant ER-Hoxb8s engrafted the paren-chyma (*Figure 6C/D*, n=5 biological replicates). Notably, both D1113H homozygous mutant brains (*Figure 6C*) and mice engrafted with D1113H ER-Hoxb8 microglia-like cells exhibited similar and persistent upregulation in brain Isg15, with pockets of robust expression around engrafted Iba1+ MLCs, clustering around Iba1- nuclei that morphologically resemble neurons (*Figure 6D*, *Figure 6—figure supplement 2A*). To get a clearer idea of their overall engraftment potential, we transplanted D1113H mutant ER-Hoxb8s into Csf1r$^{-/-}$ mice, which are unconfounded by repopulating endogenous microglia. Similar to brains transplanted with Adar1/Ifih1 dKO cells, we found dispersed cell engraft-ment that did not reach quantifiable levels (*Figure 6—figure supplement 2B*, n=3 biological repli-cates). Consistent with results from Cx3cr1CreERT; Csf1rfl/fl host mice, we saw upregulation in brain Isg15 clustering around Iba1- nuclei (*Figure 6—figure supplement 2C*, n=3 biological replicates).

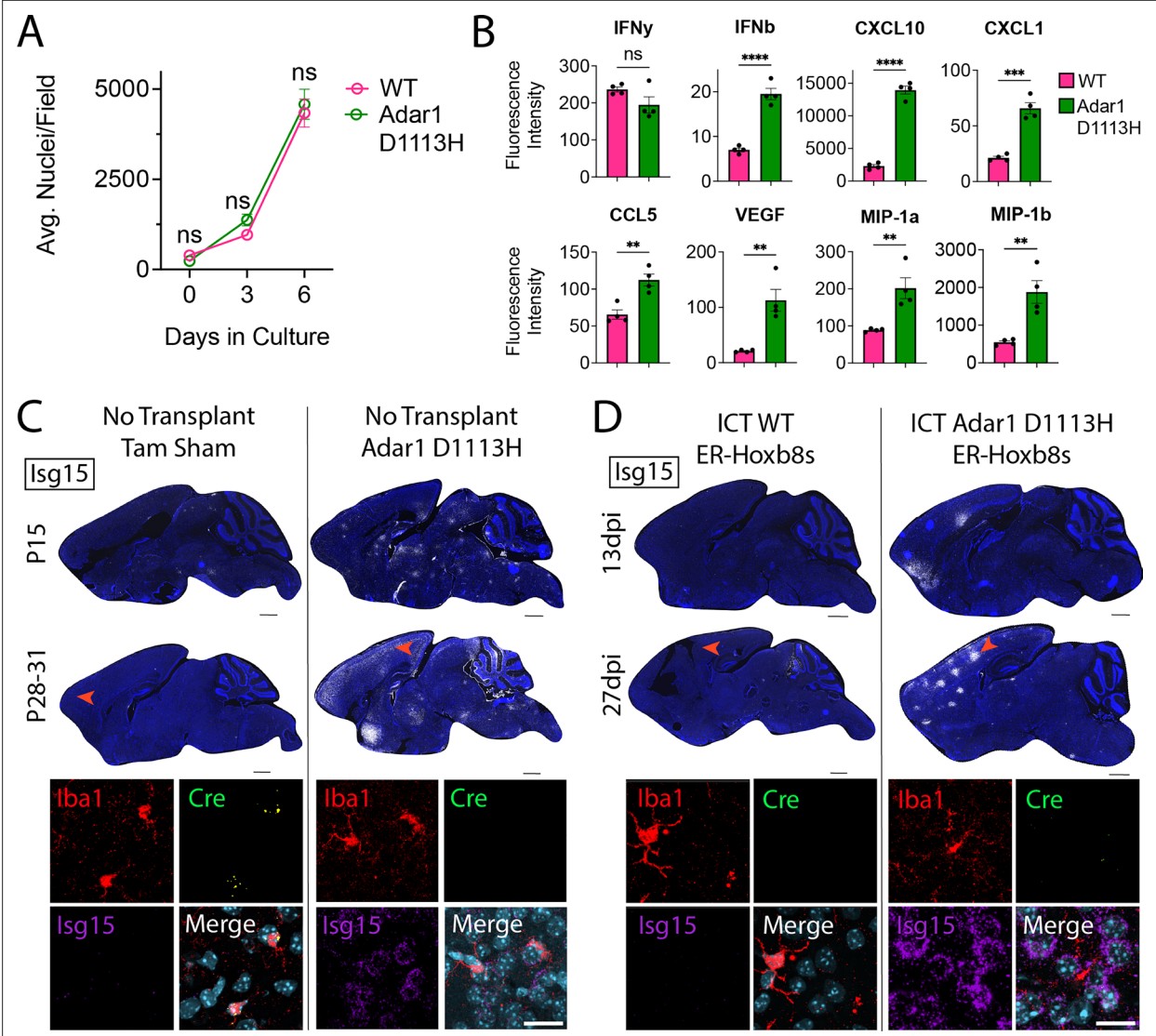

**Figure 6.** Adar1 D1113H mutant ER-Hoxb8 macrophages persistently drive brain ISG expression. (**A**) In vitro ER-Hoxb8 cell counts over time (p-values calculated via two-way ANOVA with multiple comparisons). (**B**) Multiplex bead array data for interferons, cytokines, and chemokines produced via ER-Hoxb8 macrophages (p-values calculated via independent two-sample *t*-test). (**C**) Sagittal sections of non-transplanted (tamoxifen [tam] sham control) Cx3cr1CreERT; Csf1rfl/fl brains (left) and Adar1 D1113H mutant brains (right) at age P15 and P28-31; nuclei (blue, DAPI), Isg15 (white via RNA in situ hybridization [ISH]); scale bar = 1000 um; red arrow depicts location of corresponding closeup images below, showing IBA1 (red, protein stain), Cre (green, ISH), Isg15 (purple, ISH), and nuclei (teal, DAPI); scale bar = 20 um; see *Figure 6—figure supplement 2A* for further corresponding closeup images. (**D**) Sagittal sections of Cx3cr1CreERT; Csf1rfl/fl brains intracranially transplanted with WT ER-Hoxb8s (left) and Adar1 D1113H ER-Hoxb8s (right) at 13 and 27 days post-injection (dpi); nuclei (blue, DAPI), Isg15 (white, ISH); scale bar = 1000 um; red arrow depicts location of corresponding closeup images below, showing IBA1 (red, protein stain), Cre (green, ISH), Isg15 (purple, ISH), and nuclei (teal, DAPI); scale bar = 20 um; see *Figure 6—figure supplement 2A* for further corresponding closeup images. All p-values characterized by: ns = not significant or p≥0.05, *p<0.05, **p<0.01, ***p<0.001, ****p<0.0001.

The online version of this article includes the following source data and figure supplement(s) for figure 6:

**Source data 1.** Raw numerical values for *Figure 6* plots.

**Figure supplement 1.** Evidence for Cre expression, relating to *Figure 6*.

**Figure supplement 2.** Extended Isg15 RNA in situ hybridization (ISH), relating to *Figure 6*.

In sum, D1113H ER-Hoxb8 macrophages produce type I interferon and ISGs and induce Isg15 production by neighboring cells after brain engraftment. These results demonstrate that Adar1 mutation in microglia-like cells is sufficient to drive brain ISG production.

## Discussion

Realizing the potential of microglia replacement requires new tools for research and therapy development. Here, we show the power of estrogen-regulated (ER) homeobox B8 (Hoxb8) conditionally immortalized myeloid cells to robustly model microglia for rapid assessment of gene-targeted perturbations on their identity and function. We find that ER-Hoxb8 macrophages are similar to BMD macrophages both in vitro and in vivo, where they robustly engraft the microglia-deficient brain and upregulate microglial identity genes. We then use this model to demonstrate an approach to untangle macrophage contributions to a neurological disease, establishing the impact of macrophage-specific mutations to clinically relevant phenotypes both in vitro and in vivo.

Microglia are difficult to target and manipulate for preclinical study, necessitating the use of robust microglial surrogates for discovery, such as the ER-Hoxb8 system described here. Powerfully programmed by the brain environment, microglia lose their unique transcriptional identity in vitro (*Bohlen et al., 2017*; *Gosselin et al., 2017*) and are difficult to engineer (*O'Brien et al., 2022*). Common surrogates include macrophages from isolated bone marrow or immortalized cell lines derived from macrophages or microglia (*Stanley and Heard, 1977*; *Raschke et al., 1978*; *Tsuchiya et al., 1980*; *Righi et al., 1989*; *Blasi et al., 1990*; *Chanput et al., 2014*; *Muffat et al., 2016*; *Timmerman et al., 2018*).

BMD cells are finite in number and exist as a heterogeneous population. Cell lines such as induced pluripotent stem cell-derived microglia-like cells can be expensive to generate and time-intensive to maintain (*Timmerman et al., 2018*). ER-Hoxb8 cells are expandable, readily transducible, and brain-engraftable, making them a robust microglia surrogate. Building on the work of others (*Roberts et al., 2019*; *Accarias et al., 2020*; *Xu et al., 2022*), we demonstrated straightforward gene editing using CRISPR-Cas9, generated stable lines using standard cell culture reagents, and found that ER-Hoxb8s are transcriptionally similar to primary BMD cells in vitro and in vivo. We used these strengths (and spared months of generating transgenic mouse lines) to readily test hypotheses about Adar1 function in microglia to demonstrate the potential of this system. We generated Adar1 knockout (KO) ER-Hoxb8 lines, as well as ER-Hoxb8 lines derived from mice harboring Adar1 patient mutations, to study the effect of microglia-specific Adar1 mutation on macrophage generation and interferon production. The ease with which ER-Hoxb8 cells can be modified, sorted, clonally expanded, banked, and transplanted was critical to these discoveries and will enable further innovation for the study of central nervous system macrophages.

In the future, the ER-Hoxb8 system will enable additional study of Adar1 and microglia in AGS. We found that both complete Adar1 KO, as well as an RNA-editing catalytic domain mutation in microglia-like cells, leads to increased interferon responses cell-autonomously in vitro, as well as non-cell-autonomously as demonstrated by direct transplant. This effect is blocked by Ifih1/MDA5 deletion, which is thought to sense abnormal RNA species formed by Adar1 mutation and rescues Adar1-catalytic domain point mutation phenotypes (*Liddicoat et al., 2015*; *Guo et al., 2022*). In contrast, brain engraftment is impacted by Adar1 KO regardless of Ifih1 genotype and not by Adar1 catalytic domain mutation, suggesting that the full KO phenotype may be independent of RNA editing and interferon production. Mitochondrial antiviral signaling (MAVS) adaptor protein rescues the embryonic lethality of Adar1 null mice (*Mannion et al., 2014*), suggesting that manipulation of other Adar1 pathway components could reveal Adar1's role in brain engraftment and macrophage maintenance. This system can allow for interrogation of alternative Adar1 functions, as well as other AGS mutations, and enables testing of AGS-targeted therapies.

All cell models have limitations, and we highlight two of particular relevance to the study of microglia. First, like primary BMD macrophages, ER-Hoxb8 macrophages derive from definitive as opposed to primitive hematopoiesis - the authentic origin of microglia. Though very similar to microglia, ER-Hoxb8s cannot attain full microglial identity, conspicuously lacking Sall1 expression, an important transcription factor in regulating homeostatic microglia (*Buttgereit et al., 2016*). Second, we observe that ER-Hoxb8 macrophages express slightly higher levels of inflammation-associated genes than primary BMD macrophages after brain engraftment. Though insufficient to elicit GFAP

upregulation, a sensitive biomarker for inflammation, it remains an important consideration for experimental design. Interestingly, a recent study suggests it may be possible to address both limitations by treating yolk sac progenitors with the ER-Hoxb8 virus to generate macrophages with reduced inflammatory reactivity that may serve as more authentic microglia surrogates (*Elhag et al., 2021*). It also raises the exciting potential of an analogous system for study of human, patient-derived macrophages.

More broadly, the future of engineered microglia surrogates, including ER-Hoxb8 microglia-like cells, is bright. Leveraging the advantages of ER-Hoxb8 cells, one could create controllable gene editing via an inducible Cas9 or sgRNA and perform large, pooled screens in vitro and in vivo simultaneously, using progressively advanced methods for gene targeting. We envision microglia as potent neurotherapeutics (*Bennett and Bennett, 2020*), and toward that goal, the ER-Hoxb8 model provides an ideal milieu for the testing of therapeutic cell engineering strategies, such as payload delivery, synthetic receptors, and customizable gene circuits. Here we applied two engraftment models, utilizing both Csf1r$^{-/-}$ and inducible Cx3cr1CreERT; Csf1rfl/fl host mice. Pairing the ER-Hoxb8 system with our CSF1R inhibitor-resistant receptor transplantation models will allow even more widespread adoption of these techniques (*Chadarevian et al., 2023*). Together, the combination of accessible and modifiable microglial surrogates alongside increasingly effective transplantation models holds great potential for research and therapy development.

## Materials and methods

### Mouse models

All animal studies were performed with approval from the Children's Hospital of Pennsylvania IACUC panel (#1326) in accordance with institutional and national regulations. All animals were housed in a non-barrier facility with 12 hour light/dark cycles at 23±2°C in ventilated cages with no more than five animals per cage. Animals were provided water and standard chow ad libitum. Cages and bedding were changed weekly.

Csf1r$^{-/-}$ (FVB.129x1-Csf1rtm1Ers) and Csf1r$^{+/+}$ littermate animals on the FVB background were a generous gift from Dr. Richard Stanley (Albert Einstein College of Medicine). Cx3cr1CreERT; Csf1rfl/fl mice were generated by intercrossing JAX 021212 and 021160 strains. Adar D1113H mice (p.Asp1113His) were a generous gift from Dr. Qingde Wang (University of Pittsburgh). For experiments using GFP-expressing donor cells, we backcrossed Osb-GFP (C57BL/6-Tg(CAG-EGFP)131Osb/LeySopJ [JAX 006567]) onto FVB WT mice (JAX 001800) for 22 generations. For experiments using constitutively-expressed Cas9 donor cells, we used FVB.129(B6)-Gt(ROSA)26Sortm1.1(CAG-cas9*,-EGFP)Fezh/J mice (JAX 026558).

### In vitro experiments

#### ER-Hoxb8 conditionally immortalized cell production

Immortalization of murine myeloid progenitor cells was completed as outlined in *Wang et al., 2006*. Briefly, bone marrow progenitors were isolated from the femurs and tibias of wildtype or Cas9-expressing FVB/NJ mice using a Percoll density separation (Cytiva, 17544501). Cells were cultured for three days in RPMI media (Invitrogen, 7240047) supplemented with 10 ng/mL IL3 (Peprotech, 213-13), IL6 (Peprotech, 216-16), and SCF (Peprotech, 250-03) before transduction with MSCVneo-HA-ER-Hoxb8 virus. Originally created by Dr. David Sykes (Massachusetts General Hospital), the MSCVneo-HA-ER-Hoxb8 plasmid was a generous gift from Dr. Igor Brodsky (University of Pennsylvania School of Veterinary Medicine). 24 hours after transduction, the cells were recovered and grown for 48 hours in RPMI media (Invitrogen, 7240047) supplemented with 1 mM Na-Pyruvate (Invitrogen, 11360070), 0.0005 mM beta-estradiol (Sigma Aldrich, E2758), and 10 ng/mL GM-CSF (Peprotech, 315-03). Cells were then selected for transduction with the addition of 1 mg/mL Geneticin (Thermo Fisher, 10131035) for 48 hours. Cells were then expanded and passaged every 48–72 hours at 25–50,000 cells/mL for 2 weeks, or until all non-immortalized cells were terminally differentiated or killed.

#### Bone marrow isolations

Bone marrow harvests were completed as previously described (*Bennett et al., 2016*). Briefly, femurs and tibias were dissected and flushed with 1x PBS to collect whole bone marrow. Red blood cells were then lysed with ACK Lysis Buffer (Quality Biological, 118-156-101). For experiments requiring bone

marrow progenitors, whole bone marrow samples were subsequently enriched for lineage-negative cells using the Miltenyi Direct Lineage Cell Depletion Kit (130-110-470) via MACS Column separation with LS Columns (Miltenyi, 130-042-401). For experiments requiring bone marrow monocytes, whole bone marrow samples were subsequently enriched for monocytes using the Miltenyi Monocyte Isolation Kit (130-100-629) via MACS Column separation with LS Columns.

## Macrophage differentiation

Bone marrow-derived macrophages were differentiated in vitro by plating isolated bone marrow-derived progenitors in Petri dishes at a density of 5 million cells per dish in differentiation media – DMEM media (Thermo Fisher, 10569010), 10% FBS (VWR, 89510-186), 1% Penn/Strep (Invitrogen, 15140122), supplemented with 30 ng/mL murine M-CSF (Peprotech, 315-02). Cells were differentiated for 7–9 days in a 37°C incubator with 5% $CO_2$, with media changes every 2–3 days.

ER-Hoxb8-derived macrophages were differentiated in vitro using methods adapted from *Wang et al., 2006*. Briefly, ER-Hoxb8 progenitors were plated in petri dishes at a density of 2 million cells per dish, and then differentiated in the same media and timetable as noted above.

## Phagocytosis assay

C57BL/6 WT ER-Hoxb8 cells were differentiated for 7 days as described above. The day prior to the phagocytosis assay, cells were harvested via incubation with 3 mM EDTA, resuspended in macrophage differentiation media, diluted to 25,000 cells/mL in macrophage differentiation media, and plated in separate wells of 8 Chambered Coverglass Systems (CellVis, C8-1.5H-N; 500 µL cell dilution/12,500 cells per well). The phagocytosis assay was adapted from *Williamson and Vale, 2018* and *Flannagan et al., 2014*. Briefly, 10% phosphatidylserine (PS) beads were used to mimic apoptotic cell corpses and phosphatidylcholine (PC) beads were used to mimic live cells. 10% PS beads were 4.07 µm diameter silica beads (Bangs Labs, SS05002) coated in a supported lipid bilayer composed of 10 mol.% 18:1 PS (DOPS – Avanti, 840035), 89.90 mol.% 16:0-18:1 PC (POPC – Avanti, 850457), 0.1 mol.% atto390 dye-labeled DOPE (Atto-Tec, AD 390-161). PC beads were 4.07 µm diameter silica beads (Bangs Labs, SS05002) coated in a supported lipid bilayer composed of 99.90 mol.% 16:0-18:1 PC (POPC – Avanti, 850457), 0.1 mol.% atto390 dye-labeled DOPE (Atto-Tec AD 390-161). Beads were diluted to 500,000 beads/mL in macrophage differentiation media and added to the macrophages via media change, replacing old media with 500 uL bead dilution per chamber of the 8 Chambered Coverglass Systems. Macrophages were incubated with synthetic apoptotic cell corpse mimics or synthetic live cell mimics for 60 minutes at 37°C, 5% $CO_2$. Cells were then fixed for 10 minutes at 37°C, 5% $CO_2$ in 4% paraformaldehyde (Sigma, P6148) in PBS, permeabilized for 1 minute in 0.1% TritonX-100 (Sigma, T8787) in PBS, and stained using HCS CellMask Green (Invitrogen, H32714). After staining, cells were placed under Aqua-Poly/Mount for imaging (Polysciences, 18606). Macrophages were imaged via spinning disk confocal microscopy on an inverted Nikon Ti2-E microscope fitted with a CSU-W1 spinning disk unit and Hamamatsu ORCA-FusionBT camera. Images were acquired using a 40x/0.95 NA objective in .nd2 format using Nikon Elements software and converted to ome-tif stacks for analysis in Fiji/ImageJ. Beads were considered ingested if HCS CellMask Green signal was visible fully surrounding a circular area inside the cell corresponding to the 4 µm diameter of the silica beads. Experiments were replicated four times on four different days using independently prepared reagents each day. 81–100 cells were examined per replicate condition.

## Viral production and transduction

To create retroviral supernatants, Lenti-X 293T cells (Takara Bio, 632180) were transfected using 850 ng/mL pCL-Eco (Addgene, 12371) and 850 ng/mL desired retroviral plasmid, supplemented with 5 uL/mL LipoD293 (SignaGen, 504782) in media containing DMEM (Thermo Fisher, 10569010), 25 mM HEPES (Invitrogen, 15630080), 10% FBS (VWR, 89510-186), and 1% Penn/Strep (Invitrogen, 15140122). Six hours post-transfection, media was replaced and viral supernatants were collected 48 and 72 hours later. Viral collections were combined and concentrated 10x using Retroviral Precipitation Solution (Alstem, VC200) according to manufacturer's instructions.

Transduction of ER-Hoxb8 progenitors was completed in 12-well plates coated with 1 ug/mL fibronectin (Sigma-Aldrich, F1141). Cells were plated at 200,000 cells/mL and desired titer volume of virus was added. Plates were then spun for 90 minutes at 1000 × *g*. Post-spinfection, cells were gently

mixed and resuspended in an additional 2–3 mL of fresh media. 24 hours post-transduction, cells were recovered and expanded at normal growing conditions (25–50,000 cells/mL at 37°C, 5% $CO_2$).

## CRISPR editing

Target gene guide sequences were created using the Broad Institute's CRISPick website (Mouse GRCm38 reference genome, CRISPRko mechanism, SpyoCas9 enzyme, tracrRNA) and adapted to ligate into our plasmid backbone. This backbone, which was a generous gift from Dr. Will Bailis (University of Pennsylvania; *Bailis et al., 2019*) was adapted from the MSCV-MigR1-GFP/BFP plasmid (Addgene, 27490) to include a U6-driven gRNA scaffold for the insertion of guide sequences. The BbsI-HF restriction enzyme (NEB, R3539) was used to linearize the plasmid backbone at the gRNA scaffold site, and annealed guide oligos were ligated in. The ligated plasmids were then transformed using NEB Stable Cells (C3040), plated onto LB plates supplemented with 50 ug/mL ampicillin, and grown for 24 hours at 30°C. Correct insertion of guide sequences were confirmed using bacterial colony sanger sequencing via GeneWiz/Azenta, and full plasmid sequences were confirmed using Plasmidsaurus. Viral supernatants were then created as described above and Cas9-expressing ER-Hoxb8 macrophage progenitor cells were transduced as described above. Cells were then double-FACS sorted for transduction via fluorophore expression and expanded as normal. Successful editing was confirmed using TIDE as described in *Brinkman et al., 2014*. Briefly, gDNA was collected from both control and experimental macrophages and the cut site was amplified via PCR and sent for Sanger sequencing. Indel creation was then confirmed via the TIDE application (*Brinkman et al., 2014*; http://shinyapps.datacurators.nl/tide/). When an antibody was commercially available, such as with TLR4 (Biolegend, 145403, PE), loss of protein was also confirmed via flow cytometry.

Guide oligo sequences:

Adar1 KO (sgRNA #1): GACACTGGCCAAGATTCCCA
Adar1 KO (sgRNA #2): TACCTGAACACCAACCCCGT
Ifih1 KO: TGTGGGTTTGACATAGCGCG
NTC: GCTTCCGCGGCCCGTTCAA

TIDE primer sequences:

Adar1 KO (sgRNA #1): CCATGTTGCCTTCTACAGGG, GGGCAGAGGAGTTTGACAC
Adar1 KO (sgRNA #2): CAGAGGAAGCCATGAAGG, ACACACGGTGACATTCATGC
Ifih1 KO: GCAGAGGACAGCTTCAGGA, ATAGAAGGCTAGATGCAGCGG

## TNF-a ELISA

ER-Hoxb8 progenitor cells were plated and differentiated as described in the above 'Macrophage Differentiation' methodology. After 7 days in vitro, cells were gently removed from the Petri dishes using ice cold 1x PBS + 2 mM EDTA (Thermo Fisher, 15575020). Cells were then re-plated at 50,000 cells/well in a TC-treated 96-well plate in 100 uL fresh differentiation media. 24 hours later, LPS (Sigma, L2880-25MG) and R848 (Invitrogen, TLRL-R848) were added at 100 ng/mL to respective wells. After 9.5 hours, supernatants were collected and diluted to 25% initial concentration to stay within kit ranges. TNF-a concentrations were measured using the TNF alpha Mouse Uncoated ELISA Kit with Plates (ThermoFisher, 88-7324-22) according to manufacturer's instructions.

## *Adar1* immunofluorescence

ER-Hoxb8 progenitor cells were plated and differentiated as described in the above 'Macrophage Differentiation' methodology. NTC and Adar1-edited cells were plated at 5000 cells per well in a glass-bottom 96-well plate. Cells were fixed on day eight of differentiation in 4% PFA (EMS, 15714) for 10 minutes. Cells were washed, blocked in 0.1% Tween20 (Bio-Rad, 1706531) + 5% donkey serum (Sigma, S39-100ML) for 1 hour, and then stained overnight with CD11B (BioLegend, 101208) and Hoescht 33342 Solution (Thermo Fisher, 66249) at 4°C. The following morning, cells were washed and imaged at 40x magnification Z-stacks using a BZ-X800 (Keyence) microscope.

### *Adar1* cell quantification

ER-Hoxb8 progenitor cells were plated and differentiated as described in the above 'Macrophage Differentiation' methodology. NTC and Adar1-edited cells were plated at 5000 cells per well in glass-bottom 96-well plates (n=3 independently differentiated replicates per condition). Cells were plated in fresh differentiation media supplemented with nothing, vehicle (DMSO), or 10 uM Baricitinib (Selleck Chemicals, S2851). Cells were plated with 125 uL media on day 0, 62.5 uL media was added on day 3, media was removed and replaced with 100 uL media on day 6. On differentiation days 0, 3, 6, and 9, corresponding subsets of cells were stained with 1:10,000 Hoescht for 30 minutes at 37°C and imaged with the ImageXpress Micro Confocal System. Images were taken at 10x magnification at nine sites per well, with exposure settings consistent between wells. Cell number quantification was performed using the 'Count Nuclei' pipeline in the ImageXpress software. Signal thresholds were set, masks were created over the Hoescht+ signal, and nuclei were counted (3–16 pixel width). Cell counts were averaged across the nine imaging sites and plotted as average nuclei per field per day of differentiation.

### *Adar1* multiplex arrays

ER-Hoxb8 progenitor cells were plated and differentiated as described in the above 'Macrophage Differentiation' methodology. NTC and Adar1-edited cells (Adar1 KO #1, Adar1 KO #3, Adar1/Ifih1 dKO) were plated at 100,000 cells per well in an untreated 24-well plate (n=4 independently differentiated replicates per condition), whereas WT and D1113H cells were plated at 25,000 cells per well in an untreated 24-well plate (n=4 independently differentiated replicates per condition). Media was changed every 2 days, with the baricitinib groups receiving 0 uM, 0.00064 uM, 0.16 uM, 0.4 uM, or 10 uM Baricitinib (Selleck Chemicals, S2851) with each media change. On day 7 of differentiation (24hours post final media change), cell supernatant was collected, spun down, and immediately frozen. Multiplexing analysis was performed using the Luminex 200 system by Eve Technologies Corp. (Calgary, Alberta). Assays used include the Mouse Cytokine/Chemokine 44-Plex Discovery Assay Array (MD44), the Mouse Cytokine/Chemokine 32-Plex Discovery Assay Array (MD32), as well as the Mouse IFN-a + IFN-b Assay (MDIFNAB). For the baricitinib groups, RNA was immediately collected from adherent cells for subsequent sequencing analyses.

## In vivo experiments

### Neonatal transplantation

For intracranial transplantation into Csf1r$^{-/-}$ mice, p0-p5 pups were randomly assigned and injected as previously described (*Bohlen et al., 2017*) by hand using a pulled glass capillary tube (World Precision Instruments, 1B100F-4) in an electrode holder connected by silicon tubing to a syringe. One microliter containing a single-cell suspension (300,000 cells/uL unless otherwise noted) of donor cells in 1x PBS was slowly injected bilaterally into cortex, 1–2 mm anterior and 2–3 mm lateral to lambda at a depth of 0.5–1 mm. Host animals were harvested 12–16 days after injection unless otherwise noted.

### Microglia/MLC isolations

Single-cell suspensions of microglia/engrafted brain macrophages were isolated as previously described (*Bennett et al., 2016*). Briefly, mice were anesthetized using a cocktail of ketamine (100 mg/kg) plus xylazine (10 mg/kg) in 1x PBS and perfused with 10 mL cold PBS. Brains were dounce homogenized in 10 mL cold Medium A Buffer – 10% 10x HBSS (Fisher, 14185052), 1.5% 1 M HEPES (Invitrogen, 15630080), 1.67–30% glucose (Sigma, G7021-1KG), 86.8% ddH$_2$O supplemented with 2% DNase (Worthington, LS002007). Homogenate was filtered through a 70 um strainer and pelleted. Microglia were collected following a spin with isotonic percoll – 10% 10x HBSS, 90% Percoll PLUS density gradient media (GE Healthcare, 17544501). Cells were washed with Medium A Buffer and resuspended in desired media. Mice were excluded from the study if they did not survive to the predetermined harvest timepoint.

### In vivo baricitinib treatment

Baricitinib (Selleckchem, S2851) was reconstituted in DMSO (Sigma-Aldrich, D26250) and added to a diluent of 40% PEG300 (Sigma-Aldrich, 90878), 10% Tween80 (Sigma-Aldrich, P1754), and 50% water.

Mice were administered 20 uL daily by intraperitoneal injection at doses of 10 mg/kg, 5 mg/kg, or 1 mg/kg.

## Staining experiments

### Immunofluorescence

Mice were perfused with 15 mL 1x PBS followed by 15 mL 4% PFA (EMS, 15714) and organs were then drop-fixed in 4% PFA for 16–24 hours at 4°C. Organs were then cryo-protected in 30% sucrose (Neta Scientific, SIAL-S5391-25KG), embedded in OCT (Fisher, 23-730-571), cryo-sectioned at 16 um, mounted on Superfrost Plus slides (Fisher, 1255015), and frozen at –80°C. Slides were then thawed at 60°C, rehydrated in 1x PBS, and blocked for 1 hour at RT in blocking buffer – 90% 1x PBS, 9.5% donkey serum (Sigma, S39-100ML), 0.5% Triton X-100 (Sigma, T8787-50ML). After blocking, slides were incubated with primary antibodies (described below) in staining buffer (98.5% 1x PBS, 1% donkey serum, 0.5% Triton X-100) overnight at 4°C. In the morning, slides were washed with 1x PBS and incubated with secondary antibodies (described below) for 1 hour at RT in staining buffer. Slides were washed once more and coverslipped with DAPI mounting media (EMS, 17989–60) before imaging.

Antibodies used: rabbit Anti-Iba1 (Fujifilm, 019-19741, 1:500); mouse Anti-GFAP (Agilent, MAB360, 1:500); donkey anti-rabbit IgG 594 (Thermo Fisher, A-21207, 1:500); donkey anti-mouse IgG 647 (Thermo Fisher, A32787, 1:500).

### RNA in situ hybridization

In situ hybridization (ISH) was performed on mounted fixed frozen tissues using the RNAscope system per the manufacturer's protocol (Advanced Cell Diagnostics). Fluorescent detection was achieved using the RNAscope Multiplex Fluorescent Detection kit V2 (cat # 323110). Samples were probed for mouse Isg15 (cat # 559271-C2) and Cre (cat # 312281-C3) and visualized with TSA Vivid dyes (cat #s 7534 and 7536). Following ISH, slides were immunostained with rabbit anti-Iba1 (cat # 019-19741, Fujifilm WAKO) and detected using fluorescently labeled donkey anti-rabbit secondary antibody (A21207, Invitrogen). Slides were coverslipped and nuclei were stained with Prolong Gold Antifade Mounting Medium with DAPI (cat # P36931, Invitrogen).

### Imaging acquisition and processing

Slides were imaged using a BZ-X800 (Keyence) microscope. Whole-organ tile images plus z-stack images were captured and stitched/compressed using Keyence Analyzer software prior to image export. All images within experiment panels were taken with equivalent exposure settings. All images were then analyzed with FIJI (https://imagej.net/Fiji), equivalently adjusting only for brightness and black values (raw images available upon request).

Engraftment renderings were created using GFP (488) or Iba1 (594) and DAPI channels of 10x stitches. Background was equivalently subtracted using the 'subtract background' function before manually thresholding via the 'otsu' setting. Rendered dot masks were created by analyzing particles (size = 2–144 px; circularity = 0.5–1.0) and overlaying them on the corresponding DAPI channel. The boundaries of the brains were then outlined using the 'polygon selections' tool, and everything outside this boundary was cleared and excluded from the final image.

16 um tissue sections used for RNA scope were imaged using an Andor BC43 spinning disc confocal microscope (Oxford Instruments). Sections that were directly compared were acquired using the same exposure, laser power, gain, and resolution settings. Subsequently, image analysis on raw unmanipulated images was performed using Imaris software. Representative images displayed for publication were processed equally in Imaris across all conditions to best display the data.

### Percent area quantifications

For each brain sample, average total engrafted area was calculated from three matched sagittal sections (200 uM apart starting at the medial-most point of the tissue), each using a 4x tile-scan image collected via Keyence microscope. Calculations were made in FIJI using the Iba1 and/or GFP channel tiled images. FIJI's 'polygon selections' tool and 'measure' function were used to draw and calculate a total brain ROI, as well as engrafted area ROI(s). The latter were defined as regions with Iba1+ and/or GFP+ cell groups with no fewer than 50 total cells per 'group' nor 50 cells/mm². Edge

limits of engraftment areas were defined by the nucleus of the edge-most cell being no more than 200 uM from nearest-neighbor cells. Percent area engrafted was then calculated by dividing total engrafted area(s) by total brain area. Olfactory bulb was omitted for consistency as it was not present in all samples.

For each brain sample, average total area of GFAP+ cells was calculated from three matched sagittal sections (250 uM apart starting at the medial-most point of the tissue), each using a 10x tile-scan image collected via Keyence microscope. Calculations were made in FIJI using the GFAP channel tiled images. Background was equivalently subtracted from all images using the 'subtract background' function. Images were then converted into 8-bit format and thresholded at 40/255 with the 'intermodes' setting. ROIs were manually drawn for cortex, hippocampus, midbrain, and thalamus using FIJI's 'polygon selections' tool. Percent area covered was then calculated using the 'measure' function to quantify positive GFAP signal over total ROI area.

## Cell density quantification

For each brain sample, average cortical cell density was calculated using the same three matched sagittal sections used above to calculate area of engraftment. Here, three 20x representative images were included that spanned the cortical region of engraftment. If the region of engraftment was small, fewer images were included such that no cells overlapped and were counted more than once. Cells were counted manually using FIJI's 'Cell Counter' tool and were defined as cells that were Iba1+ and/or GFP+ and directly overlaid with a DAPI+ nucleus. Cell counts per image were averaged and divided by total area per image to obtain densities at cells/mm$^2$.

## Morphological analysis

All morphological analyses were performed using Fiji. Individual Iba1+ cells in the cortical parenchyma were randomly selected from 10x stitched images of the brain. For all images, individual cells were converted to 8-bit and thresholded equivalently. The despeckle function was used to remove noise from each sample. Each image was then converted to a binary image that was used for further analysis. Soma size was measured by drawing a region of interest (ROI) around somas and measuring the encompassed area. Primary branches were defined as branches extending directly from the soma and were counted manually. The length of each branch was measured from the center of the cell soma using the Neuroanatomy SNT plugin (*Arshadi et al., 2021*). Traces were drawn to outline branch morphology, then measured using the SNT Measurements function to obtain branch lengths.

## Flow cytometry and FACS

Once isolated into single cell suspensions, cells were washed 2x in FACS Buffer - 1x PBS, 0.5% BSA (Sigma, 126609-10GM), 0.5 mM EDTA (Thermo Fisher, 15575020) – and blocked with FC Block – 0.2% CD16/CD32 (BD Biosciences, 553142) in FACS Buffer – for 10 minutes at RT. Cells were then stained using antibodies (described below) for 30–60 minutes at 4°C. Cells were washed 2x more in FACS Buffer before being analyzed or sorted as described below.

Antibodies used: LIVE/DEAD fixable far red dead cell stain kit (Invitrogen, L10120, 1:1600); anti-mouse CD45 (PE/Cyanine7, Clone 30-F11, Biolegend, 103113, 1:400); anti-mouse/human CD11b (PerCP/Cyanine5.5, Clone M1/70, Biolegend, 101227, 1:400); anti-mouse monoclonal TMEM119 (PE, Clone V3RT1GOsz, Invitrogen, 12-6119-82, 1:400).

For flow cytometry experiments, stained cells were analyzed on a CytoFLEX LX (6 Laser) Flow Cytometer and visualized via FlowJo (10.8.1). For sorting experiments, stained cells were sorted with a 100 uM nozzle on a FACSAria Fusion, FACSJazz, Aurora, or MoFlo Astrios depending on Core availability (no effects on data were seen between FACS machines). Cells were deposited into media for cell expansion, or TRIzol LS Reagent (Fisher, 10296028) for subsequent RNA extraction.

## Bulk RNA-sequencing and analysis

### RNA extraction and quantification

For in vitro populations, RNA was extracted using the Qiagen RNeasy Mini Kit (74104) according to manufacturer's instructions. For suspended populations, cells were collected and spun down at 200 × *g* for 5 minutes, supernatant was removed, and 350 uL Buffer RLT was added directly to the cell pellet.

For adherent populations, supernatant was removed, plates were washed with 1x PBS, and 650 uL of Buffer RLT was added directly to the dish to create cell lysates.

For in vivo populations, cells were isolated as described in the above 'Microglia/MLC Isolations' methodology and FACS sorted into TRIZol LS reagent as described in the above 'Flow Cytometry and FACS' methodology. Cells were briefly vortexed and 0.2 volumes of chloroform (Sigma, C2432-500ML) were added. Cells were again vortexed, incubated for 3 minutes at RT, and then centrifuged at 12,000 x $g$ for 15 minutes at 4°C. The upper aqueous phase was then collected and 1 volume of fresh 70% EtOH was added. Cells were again vortexed and then run through the Qiagen RNeasy Micro Kit (74004) according to manufacturer's instructions. Isolated RNA was stored at –80°C.

When able, RNA Integrity Numbers (RINs) were calculated using an Agilent TapeStation with High Sensitivity RNA ScreenTape (5067-5579) and Sample Buffer (5067-5580) according to manufacturer's instructions. Cells with a RIN $\geq$ 7.5 were used for subsequent library preparations. For in vitro datasets, the average RIN score was 9.03. Quantification was often not possible for brain-isolated cell populations due to low yield. In this case, all samples were used for subsequent library preparations, and quality was assessed later during preparation.

## Library preparation and sequencing

Sequencing libraries were prepared in-house using either 40 ng or 8 uL RNA per sample, depending on if RIN scores and concentrations were able to be calculated as discussed above, using the NEBNext Single Cell/Low Input RNA Library Prep Kit for Illumina (NEB, E6420) according to manufacturer's instructions. Quality and quantity steps were performed using an Agilent TapeStation using High Sensitivity D5000 ScreenTape (5067-5592), and High Sensitivity D5000 Reagents (5067*-5593).

Bulk RNA-sequencing of completed libraries was performed by the Children's Hospital of Philadelphia's Center for Applied Genomics (CAG) Core Facility using a NovaSeq 6000 (Illumina) system with the SP Reagent Kit v1.5 (2 × 100 bp). Data was de-multiplexed and sent to us as FastQ files via BaseSpace (Illumina).

For sequencing of NTC and Adar1-edited cells with varying doses of Baricitinib, cDNA libraries were sent to Novogene for sequencing using a NovaSeq 6000 (Illumina) system with the S4 Reagent Kit (2 × 150 bp). Data was de-multiplexed and sent to us as FastQ files for manual download. Data sequenced by Novogene was never combined with data sequenced by CAG.

## Sequencing analysis

All pre-processing steps were run using Terminal. FastQ files were downloaded from BaseSpace (Illumina) and concatenated across lanes using the 'cat' function. All files were processed using FastP (*Chen et al., 2018*; *Chen, 2023*) to filter reads and trim Illumina adaptors, as well as FastQC (*Andrews, 2014*) to assess quality. All files were determined to be of sufficient quality for subsequent downstream processing. Reads were pseudoaligned to the mouse GRCm39 cDNA transcriptome (Ensembl release 99; *Schneider et al., 2017*; *Zerbino et al., 2018*) using Kallisto (version 0.46.0; *Bray et al., 2016*) and run through MultiQC (*Ewels et al., 2016*) for a final quality check after mapping.

All post-processing steps were run using RStudio (*R Development Core Team, 2022*). Transcripts were summarized to the gene level through tximport (version 1.14.2; *Soneson et al., 2015*) with abundance counts calculated via TPM and then normalized using TMM via edgeR (*Robinson et al., 2010*; *McCarthy et al., 2012*; *Chen et al., 2016*). Hierarchical cluster dendrograms were created using the 'dist'function (method = euclidean) and 'hclust' function (method = complete) and were visualized with the plot function. PCA plots were created using the 'prcomp' function. Differential gene expression analysis was performed using linear modeling via limma (*Ritchie et al., 2015*), and differentially expressed genes (DEGs) were decided by applying an FDR cutoff of 0.05, a counts per million (CPM) cutoff of +1, and a Log2(FC) cutoff of ±2. Linear models and $R^2$ values were created using the 'lm' function. All output plots were visualized with ggplot2 (*Wickham, 2016*).

Statistical overrepresentation tests and related GO Terms were calculated using the Panther Classification System (version 18.0). Respective lists of DEGs were manually imported into the Gene List Analysis tab, selected for *Mus musculus*, and run through the statistical overrepresentation test option using the PANTHER GO-Slim options for biological process, molecular function, and cellular component. Tests were run using the Fisher's Exact option while calculating the false discovery rate. FDR was set to a threshold of 0.05, and terms were ranked via the hierarchy option.

## Statistical calculations

GraphPad Prism (version 10.0.0) was used to perform statistical tests and generate p values with standard designations (ns = not significant or $p \geq 0.05$, *$p<0.05$, **$p<0.01$, ***$p<0.001$, ****$p<0.0001$). All values are shown as mean ± standard error of the mean (SEM). Details regarding replicate numbers and individual statistical test used are provided in the respective figure legends. No explicit power analyses were performed, but group size was based on previous studies using similar approaches. Blinding was used whenever possible, such as for all cell quantifications, morphology analyses, and multiplex arrays.

## Acknowledgements

We thank Dr. Richard Stanley (Albert Einstein College of Medicine, New York, NY, USA) for the Csf1r$^{-/-}$ (FVB.129X1-Csf1rtm1Ers) and Csf1r$^{+/+}$ littermate animals on the FVB background; Dr. Qingde Wang (University of Pittsburgh, Pittsburgh, PA, USA) for the Adar1 D1113H mice (p.Asp1113His); Dr. David Sykes (Mass General Hospital, Boston, MA, USA) and Dr. Igor Brodsky (University of Pennsylvania, Philadelphia, PA, USA) for the MSCVneo-HA-ER-HoxB8 plasmid; Dr. Dan Beiting for the bulk RNA sequencing DIY Transcriptomics scripts; the Children's Hospital of Philadelphia Center for Applied Genomics for bulk RNA sequencing assistance; the Children's Hospital of Philadelphia Flow Cytometry core for flow cytometry and sorting assistance; https://www.biorender.com/ for graphics; and Kayla Peelman (Emory University, Atlanta, GA, USA) for assistance with Illustrator. This work was supported by National Science Foundation Graduate Research Fellowship Program DGE-1845298 (KMN, SIL); NIH Training in Age Related Neurodegenerative Diseases T32 2-T32-AG-000255–26 (VSC); NIH Medical Scientist Training Program T32 GM007170 (VSC); NIH T32MH019112 (WHA); NIH T32 GM008076 (SIL); Blavatnik Family Fellowship, Blavatnik Foundation (SIL); NIH R35GM138085 (WB); Paul Allen Institute Distinguished Investigator Award (WB); NIH R15NS133939 (APW); R01AI139544 (QW); R01NS134651 (QW); NIH DP5OD036159 (MLB); NIH R01-NS-120960–01 (FCB); Klingenstein-Simons Fellowship in Neuroscience (FCB); and The Paul Allen Frontiers Group GRT-00000774 (FCB). Since the submission of this manuscript, we lost our colleague and dear friend, Dr. Qingde Wang, a leader in the field of ADAR research. We wish to acknowledge his important contributions to this work, his generous mentorship of our trainees, and his lasting impact on the field. He is deeply missed.

## Additional information

### Funding

| Funder | Grant reference number | Author |
| --- | --- | --- |
| National Science Foundation | DGE-1845298 | Kelsey M Nemec, Sonia I Lombroso |
| NIH Medical Scientist Training Program | T32 GM007170 | V Sai Chaluvadi |
| National Institute of Neurological Disorders and Stroke | T32-AG-000255-26 | V Sai Chaluvadi |
| National Institute of Mental Health | T32MH019112 | William H Aisenberg |
| National Institute of General Medical Sciences | NIH T32 GM008076 | Sonia I Lombroso |
| Blavatnik Family Foundation | Blavatnik Family Fellowship | Sonia I Lombroso |
| National Institute of Allergy and Infectious Diseases | R01AI139544 | Qingde Wang |
| National Institute of Neurological Disorders and Stroke | R01NS134651 | Qingde Wang |

| Funder | Grant reference number | Author |
| --- | --- | --- |
| NIH Office of the Director | DP5OD036159 | Mariko L Bennett |
| National Institute of Neurological Disorders and Stroke | R01NS120960 | F Chris Bennett |
| Esther A. and Joseph Klingenstein Fund | Fellowship Award in Neuroscience | F Chris Bennett |
| Paul Allen Frontiers Group | | Will Bailis |
| National Institute of Health | R35GM138085 | Will Bailis |
| National Institute of Health | R15NS133939 | Adam P Williamson |
| The Paul Allen Frontiers Group | GRT-00000774 | F Chris Bennett |

The funders had no role in study design, data collection and interpretation, or the decision to submit the work for publication.

## Author contributions

Kelsey M Nemec, Conceptualization, Data curation, Formal analysis, Validation, Investigation, Visualization, Methodology, Writing – original draft, Writing – review and editing; Genevieve Uy, Freddy S Purnell, Bilal Elfayoumi, Sonia I Lombroso, Xinfeng Guo, Niklas Blank, Chet Huan Oon, Fazeela Yaqoob, Brian Temsamrit, Investigation; V Sai Chaluvadi, Formal analysis, Supervision, Investigation; Leila Byerly, Formal analysis, Investigation; Micaela L O'Reilly, Carleigh A O'Brien, William H Aisenberg, Supervision, Investigation; Priyanka Rawat, Formal analysis, Visualization; Christoph A Thaiss, Resources, Supervision; Will Bailis, Qingde Wang, Conceptualization, Resources, Supervision; Adam P Williamson, Conceptualization, Resources, Formal analysis, Supervision, Validation, Investigation; Mariko L Bennett, Conceptualization, Resources, Supervision, Funding acquisition, Writing – original draft, Writing – review and editing; F Chris Bennett, Conceptualization, Resources, Supervision, Funding acquisition, Writing – original draft, Project administration, Writing – review and editing

## Author ORCIDs

Kelsey M Nemec ⓘ https://orcid.org/0000-0003-3036-254X
Carleigh A O'Brien ⓘ https://orcid.org/0000-0003-4549-0640
Will Bailis ⓘ https://orcid.org/0000-0001-9420-6250
Qingde Wang ⓘ https://orcid.org/0000-0002-2482-7972
F Chris Bennett ⓘ https://orcid.org/0000-0003-2570-0620

## Ethics

All animal studies were performed with approval from the Children's Hospital of Pennsylvania IACUC panel (#1326) in accordance with institutional and national regulations. All animals were housed in a non-barrier facility with 12-hour light/dark cycles at 23+/-2 degrees C in ventilated cages with no more than five animals per cage. Animals were provided water and standard chow ad libitum. Cages and bedding were changed weekly.

Reviewer #1 (Public review): https://doi.org/10.7554/eLife.102900.3.sa1
Reviewer #2 (Public review): https://doi.org/10.7554/eLife.102900.3.sa2
Author response https://doi.org/10.7554/eLife.102900.3.sa3

# Additional files

## Supplementary files

MDAR checklist

Supplementary file 1. Spreadsheet of RNA-sequencing data and GO Terms.

Supplementary file 2. Raw microscopy images used for rendered tile stitches, RNAScope, and IHC/ICC panels.

## Data availability

All data used to support the claims of this study are included in the manuscript and supporting files. Source data files have been provided for all figures. Raw and processed files of bulk RNA-sequencing data have been deposited at Gene Expression Omnibus under accession GSE314288. Sequencing data used from *Bennett et al., 2018* and *Cronk et al., 2018* are publicly available at their corresponding citations.

The following dataset was generated:

| Author(s) | Year | Dataset title | Dataset URL | Database and Identifier |
|---|---|---|---|---|
| Kelsey MN, Genevieve U, Freddy SP, Bilal E, Leila B, Micaela LO, Carleigh AO, William HA, Sonia IL, Xinfeng G, Niklas B, Chet Huan O, Fazeela Y, Brian T, Priyanka R, Christoph AT, Will B, Adam PW, Qingde W, Mariko LB, Chaluvadi VS | 2025 | Microglia replacement by ER-Hoxb8 conditionally immortalized macrophages provides insight into Aicardi-Goutières Syndrome neuropathology | https://www.ncbi.nlm.nih.gov/geo/query/acc.cgi?acc=GSE314288 | NCBI Gene Expression Omnibus, GSE314288 |

The following previously published datasets were used:

| Author(s) | Year | Dataset title | Dataset URL | Database and Identifier |
|---|---|---|---|---|
| Cronk JC, Filiano AJ, Louveau A, Marin I, March R, Ji E, Goldman DH, Smirnov I, Geraci N, Acton S, Overall CC, Kipnis J | 2018 | Peripherally derived macrophages can engraft the brain independent of irradiation and maintain an identity distinct from microglia | https://www.ncbi.nlm.nih.gov/geo/query/acc.cgi?acc=GSE84819 | NCBI Gene Expression Omnibus, GSE84819 |
| Cronk JC, Filiano AJ, Louveau A, Marin I, March R, Ji E, Goldman DH, Smirnov I, Geraci N, Acton S, Overall CC, Kipnis J | 2018 | Peripherally derived macrophages can engraft the brain independent of irradiation and maintain an identity distinct from microglia [GFP] | https://www.ncbi.nlm.nih.gov/geo/query/acc.cgi?acc=GSE108569 | NCBI Gene Expression Omnibus, GSE108569 |
| Cronk JC, Filiano AJ, Louveau A, Marin I, March R, Ji E, Goldman DH, Smirnov I, Geraci N, Acton S, Overall CC, Kipnis J | 2018 | Peripherally derived macrophages can engraft the brain independent of irradiation and maintain an identity distinct from microglia [LPS] | https://www.ncbi.nlm.nih.gov/geo/query/acc.cgi?acc=GSE108575 | NCBI Gene Expression Omnibus, GSE108575 |
| Bennett ML | 2018 | A combination of ontogeny and environment establishes microglial identity | https://www.ncbi.nlm.nih.gov/bioproject/?term=PRJNA453419 | NCBI BioProject, PRJNA453419 |

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
