## [Editor Report · eLife Assessment]

This revised study describes an **important** new model for in vivo manipulation of microglia, exploring how mutations in the Adar1 gene within microglia contribute to Aicardi-Goutières Syndome. The methodology is validated with **exceptional** data, supporting the authors' conclusions. The paper underscores both the advantages and limitations of using transplanted cells as a surrogate for microglia, making it a resource that is of value for biologists studying macrophages and microglia.

---

## [Referee Report · Reviewer #1 (Public review)]

Summary:

Aicardi-Goutières Syndrome (AGS) is a genetic disorder that primarily affects the brain and immune system through excessive interferon production. The authors sought to investigate the role of microglia in AGS by first developing bone-marrow-derived progenitors in vitro that carry the estrogen-regulated (ER) Hoxb8 cassette, allowing them to expand indefinitely in the presence of estrogen and differentiate into macrophages when estrogen is removed. When injected into the brains of Csf1r-/- mice, which lack microglia, these cells engraft and resemble wild-type (WT) microglia in transcriptional and morphological characteristics, although they lack Sall1 expression. The authors then generated CRISPR-Cas9 Adar1 knockout (KO) ER-Hoxb8 macrophages, which exhibited increased production of inflammatory cytokines and upregulation of interferon-related genes. This phenotype could be rescued using a Jak-Stat inhibitor or by concurrently mutating Ifih1 (Mda5). However, these Adar1-KO macrophages fail to successfully engraft in the brain of both Csf1r-/- and Cx3cr1-creERT2:Csf1rfl/fl mice. To overcome this, the authors used a mouse model with a patient-specific Adar1 mutation (Adar1 D1113H) to derive ER-Hoxb8 bone marrow progenitors and macrophages. They discovered that Adar1 D1113H ER-Hoxb8 macrophages successfully engraft the brain, although at lower levels than WT-derived ER-Hoxb8 macrophages, leading to increased production of Isg15 by neighboring cells. These findings shed new light on the role of microglia in AGS pathology.

Strengths:

The authors convincingly demonstrate that ER-Hoxb8 differentiated macrophages are transcriptionally and morphologically similar to bone marrow-derived macrophages. They also show evidence that when engrafted in vivo, ER-Hoxb8 microglia are transcriptomically similar to WT microglia. Furthermore, ER-Hoxb8 macrophages engraft the Csf1r-/- brain with high efficiency and rapidly (2 weeks), showing a homogenous distribution. The authors also effectively use CRISPR-Cas9 to knock out TLR4 in these cells with little to no effect on their engraftment in vivo, confirming their potential as a model for genetic manipulation and in vivo microglia replacement.

Overall, this paper demonstrates an innovative approach to manipulating microglia using ER-Hoxb8 cells as surrogates. The authors present convincing evidence of the model's efficacy and potential for broader application in microglial research, given its ease of production and rapid brain engraftment potential in microglia-deficient mice. Using mouse-derived cells for transplantation reduces complications that can come with the use of human cell lines, highlighting the utility of this system for research in mouse models.

---

## [Referee Report · Reviewer #2 (Public review)]

Summary:

Microglia have been implicated in brain development, homeostasis, and diseases. "Microglia replacement" has gain tractions in recent years, using primary microglia, bone marrow or blood-derived myeloid cells, or human iPSC-induced microglia. Here, the authors extended their previous work in the area and provide evidence to support: (1) Estrogen-regulated (ER) homeobox B8 (Hoxb8) conditionally immortalized macrophages from bone marrow can serve as stable, genetically manipulated cell lines. These cells are highly comparable to primary bone marrow-derived (BMD) macrophages in vitro, and, when transplanted into a microglia-free brain, engraft the parenchyma and differentiate into microglia-like cells (MLCs). Taking advantage of this model system, the authors created stable, Adar1-mutated ER-Hoxb8 lines using CRISPR-Cas9 to study the intrinsic contribution of macrophages to Aicardi-Goutières Syndrome (AGS) disease mechanism.

Strengths:

The studies are carefully designed and well-conducted. The imaging data and gene expression analysis are carried out at a high level of technical competences and the studies provide strong evidence that ER-Hoxb8 immortalized macrophages from bone marrow are a reasonable source for "microglia replacement" exercise. The findings are clearly presented, and the main message will be of general interest to the neuroscience and microglia communities.

---

## [Author Response]

The following is the authors’ response to the original reviews

**Public Reviews:**

**Reviewer #1 (Public review):**
Summary:Aicardi-Goutières Syndrome (AGS) is a genetic disorder that primarily affects the brain and immune system through excessive interferon production. The authors sought to investigate the role of microglia in AGS by first developing bone-marrow-derived progenitors in vitro that carry the estrogen-regulated (ER) Hoxb8 cassette, allowing them to expand indefinitely in the presence of estrogen and differentiate into macrophages when estrogen is removed. When injected into the brains of Csf1r-/- mice, which lack microglia, these cells engraft and resemble wild-type (WT) microglia in transcriptional and morphological characteristics, although they lack Sall1 expression. The authors then generated CRISPR-Cas9 Adar1 knockout (KO) ER-Hoxb8 macrophages, which exhibited increased production of inflammatory cytokines and upregulation of interferon-related genes. This phenotype could be rescued using a Jak-Stat inhibitor or by concurrently mutating Ifih1 (Mda5). However, these Adar1-KO macrophages fail to successfully engraft in the brain of both Csf1r-/- and Cx3cr1-creERT2:Csf1rfl/fl mice. To overcome this, the authors used a mouse model with a patient-specific Adar1 mutation (Adar1 D1113H) to derive ER-Hoxb8 bone marrow progenitors and macrophages. They discovered that Adar1 D1113H ER-Hoxb8 macrophages successfully engraft the brain, although at lower levels than WT-derived ER-Hoxb8 macrophages, leading to increased production of Isg15 by neighboring cells. These findings shed new light on the role of microglia in AGS pathology.Strengths:The authors convincingly demonstrate that ER-Hoxb8 differentiated macrophages are transcriptionally and morphologically similar to bone marrow-derived macrophages. They also show evidence that when engrafted in vivo, ER-Hoxb8 microglia are transcriptomically similar to WT microglia. Furthermore, ER-Hoxb8 macrophages engraft the Csf1r-/- brain with high efficiency and rapidly (2 weeks), showing a homogenous distribution. The authors also effectively use CRISPR-Cas9 to knock out TLR4 in these cells with little to no effect on their engraftment in vivo, confirming their potential as a model for genetic manipulation and in vivo microglia replacement.Weaknesses:The robust data showing the quality of this model at the transcriptomic level can be strengthened with confirmation at protein and functional levels. The authors were unable to investigate the effects of Adar1-KO using ER-Hoxb8 cells and instead had to rely on a mouse model with a patient-specific Adar1 mutation (Adar1 D1113H). Additionally, ER-Hoxb8-derived microglia do not express Sall1, a key marker of microglia, which limits their fidelity as a full microglial replacement, as has been rightfully pointed out in the discussion.Overall, this paper demonstrates an innovative approach to manipulating microglia using ER-Hoxb8 cells as surrogates. The authors present convincing evidence of the model's efficacy and potential for broader application in microglial research, given its ease of production and rapid brain engraftment potential in microglia-deficient mice. While Adar1-KO macrophages do not engraft well, the success of TLR4-KO line highlights the model's potential for investigating other genes. Using mouse-derived cells for transplantation reduces complications that can come with the use of human cell lines, highlighting the utility of this system for research in mouse models.

Thank you for this thoughtful and balanced assessment. The major suggestion from Reviewer 1 was that confirmation of RNAseq data with protein or functional studies would add strength. We provided protein staining by IHC for IBA1 in vivo, as well as protein staining by FACS for CD11B, CD45, and TMEM119 in vitro and in vivo. For TLR4, we showed successful protein KO and blunted response to LPS (a TLR4 ligand) challenge, which we believe provides some protein and functional data to support the approach. To bolster these data, we added staining for P2RY12 on brain-engrafted ER-Hoxb8s.

Regarding the *Adar1* KO phenotypes showing non-engraftment. Because ADAR1 KO mice are embryonically lethal due to hematopoietic failure, we see the health impacts of *Adar1* KO on ER-Hoxb8s as a strength of the transplantation model, enabling the assessment of ADAR1 global function in macrophages and microglia-like cells without generation of a transgenic mouse line. In addition, it was a surprise that the health impact occurs at the macrophage and not the progenitor stage, perhaps providing insight for future studies of ADAR1’s role in hematopoiesis. Instead, we were able to show a significant impact of complete loss of *Adar1* on survival and engraftment, suggesting an important biological function of ADAR1. Macrophage-specific D1113H mutation, which affects part of the deaminase domain, shows that when the RNA deamination (but not the RNA binding) function of ADAR1 is disrupted, we find brain-wide interferonopathy. This is very exciting to our group and hopefully the community as astrocytes are thought to be a major driver of brain interferonopathy in patients with ADAR1 mutations. Instead, this suggests that disruption of brain macrophages is also a major contributor.

**Reviewer #2 (Public review):**
Summary:Microglia have been implicated in brain development, homeostasis, and diseases. "Microglia replacement" has gained traction in recent years, using primary microglia, bone marrow or blood-derived myeloid cells, or human iPSC-induced microglia. Here, the authors extended their previous work in the area and provided evidence to support: (1)Estrogen-regulated (ER) homeobox B8 (Hoxb8) conditionally immortalized macrophages from bone marrow can serve as stable, genetically manipulated cell lines. These cells are highly comparable to primary bone marrow-derived (BMD) macrophages in vitro, and, when transplanted into a microglia-free brain, engraft the parenchyma and differentiate into microglia-like cells (MLCs). Taking advantage of this model system, the authors created stable, Adar1-mutated ER-Hoxb8 lines using CRISPR-Cas9 to study the intrinsic contribution of macrophages to the Aicardi-Goutières Syndrome (AGS) disease mechanism.Strengths:The studies are carefully designed and well-conducted. The imaging data and gene expression analysis are carried out at a high level of technical competence and the studies provide strong evidence that ER-Hoxb8 immortalized macrophages from bone marrow are a reasonable source for "microglia replacement" exercise. The findings are clearly presented, and the main message will be of general interest to the neuroscience and microglia communities.
**Recommendations for the authors:**

**Reviewer #1 (Recommendations for the authors):**
This is an elegant study, demonstrating both the utility and limitations of ER-Hoxb8 technology as a surrogate model for microglia in vivo. The manuscript is well-designed and clearly written, but authors should consider the following suggestions:(1) Validation of RNA hits at the protein level: To strengthen the comparison between ER-Hoxb8 macrophages and WT bone marrow-derived macrophages, validating several RNA hits at the protein level would be beneficial. As many of these hits are surface markers, flow cytometry could be employed for confirmation (e.g., Figure 1D, Figure 3E).

In vitro, we show protein levels by flow cytometry for CD11B (ITGAM) and CD45 (PTPRC; Figure 1C), as well as TMEM119 (Supplemental Figure 2A) and TLR4 (Supplemental Figure 3C/D). In vivo, we show TMEM119 protein levels by flow cytometry (Figure 3A), as well as their CD11B/CD45 pregates (Supplemental Figure 2C), plus immunostaining for IBA1 (AIF1; Figure 2D). We now provide additional data showing P2RY12 immunostaining in brain-engrafted cells (Supplemental Figure 2B).

(2) The authors should consider testing the phagocytic capacity of ER-Hoxb8-derived macrophages to further validate their functionality.

Thank you for the suggestion. We measured ER-Hoxb8 macrophage ability to engulf phosphatidylserine-coated beads that mimic apoptotic cells, compared with phosphatidylcholine-coated beads, now as new Supplemental Figure 1C/D. This agrees with existing literature showing efficient engulfment/phagocytosis by ER-Hoxb8-derived cells (Elhag et al., 2021).

(3) For Figure 3E, incorporating a wild-type (WT) microglia reference would be beneficial to establish a baseline for comparison (e.g. including WT microglia data in the graph or performing a ratio analysis against WT expression levels).

We agree - we now include bars representing our sequenced primary microglia data in Figure 3E as a comparison.

(4) Some statistical analyses may require refinement. Specifically, for Figure 4J, where the effects of Adar1 KO and Adar1 KO with Bari are compared, it would be more appropriate to use a two-way ANOVA.

Thank you for noting it. We have now done more appropriate two-way ANOVA and included the updated results in Figure 4J and the corresponding Supplemental Figure 4G. Errors in figure legend texts have also been corrected to reflect the statistical tests used.

(5) Cx3cr1-creERT2 pups injected with tamoxifen: The authors could clarify the depletion ratio in these experiments before the engraftment and assess whether the depletion is global or regional. In comparison to Csf1r-/-, where TLR4-KO ER-Hoxb8 engraft globally, in Cx3cr1-creERT2, the engraftment seems more regional (Figure 5A vs Supplementary Figure 5B); is this due to the differences in depletion efficiency?

This is an excellent question and observation, and one that we are very interested in, though that finding does not change the conclusions of this particular study. We find some region-specific differences in depletion early after tamoxifen injection, but that all brain regions are >95% depleted by P7. For instance, in a recently published manuscript (Bastos et al., 2025) we find some differences in the depletion kinetics in the genetic model. By P3, we find 90% depletion in cortex with 50-60% in thalamus and hippocampus. In other studies, we typically deliver primary monocytes, and this is the first study where we report engraftment of ER-Hoxb8 cells in the inducible model. In this sense, it is possible that depletion kinetics may regionally affect engraftment, but future studies are required to more finely assess this point with ER-Hoxb8s, as it may change how these models are used in the future.

Bastos et al., Monocytes can efficiently replace all brain macrophages and fetal liver monocytes can generate bonafide SALL1+ microglia, Immunity (2025), https://doi.org/10.1016/j.immuni.2025.04.006

(6) It would be helpful for the authors to clarify whether Adar1 is predominantly expressed by microglia, especially since the study aims to show its role in dampening the interferon response.

That’s a wonderful point. *Adar1* is expressed by all brain cells, with highest transcript level in some neurons, astrocytes, and oligodendrocytes. It is an interferon-stimulated gene, and mutation itself leads to interferonopathy, we believe, due to poor RNA editing and detection of endogenous RNA as non-self by MDA5. We hope it can dampen the interferon response, but in the case of mutation, *Adar1* is probably causal of interferonopathy. It is induced in microglia upon systemic inflammatory challenge (LPS). We have edited the text to highlight its expression pattern. See BrainRNAseq.org (Zhang*, Chen*, Sloan*, et al., 2014 and Bennett et al., 2016)

**Reviewer #2 (Recommendations for the authors):**
(1) There appears to be a morphological difference between wt and Adar1/Ifih1 double KO (dKO) cells in the engrafted brains (Figure 5). It would be good if the authors could systematically compare the morphology (e.g., soma size, number, and length of branches) of the engrafted MLCs between the wt and mutant cells.

We agree. While cells did not differ in branch number or length, engrafted dKO cells had significantly larger somas compared with controls, which we now present in Figure S5A.

(2) To fully appreciate the extent of how those engrafted ER-Hoxb8 immortalized macrophages resemble primary, engrafted yolk sac-myeloid cells, vs engrafted iPSC-induced microglia, it would be informative to provide a comparison of their RNAseq data derived from the engrafted ER-Hoxb8 immortalized macrophages with published data transcriptomic data sets (e.g. Bennett et al. Neuron 2018; Chadarevian et al. Neuron 2024; Schafer et al. Cell 2023).

Thank you for this suggestion. To address this, we provide our full dataset for additional experiments. To compare with a similar non-immortalized model, we compared top up- and down-regulated genes from our data to those of ICT yolk sac progenitor cells from our previous work (Bennett et al., 2018). We find overlap between brain-engrafted ER-Hoxb8-, bone marrow-, and yolk sac-derived cells (Supplemental Figure 2F, Supplemental Table 3).

Minor comments:Figure 6C: red arrow showing zoom in regions are not matchable. It might be beneficial to provide bigger images with each channel for C and D as a Supplemental Figure.

We fixed this in Figure 6C to show areas of interest in the cortex for both conditions. Figure S7A shows intermediate power images to aid in interpretation.